# Balancing Interpretability and Accuracy: Energy-Ensemble Concept Bottleneck Models for Enhanced Concept Inference

## Abstract

Concept bottleneck models (CBM) have emerged as a promising solution to address the lack of interpretability in deep learning models. However, recent researches on CBM prioritize task accuracy at the expense of interpretability, weakening their ability to accurately infer key concepts. This work addresses this trade-off by introducing the energy ensemble CBM (EE-CBM). The EE-CBM leverages an energy-based concept encoder to effectively extract concepts, overcoming the information bottleneck common in conventional CBMs. Additionally, a novel energy ensemble gate within the EE-CBM architecture efficiently combines energy and concept probability to further address this bottleneck. Moreover, the EE-CBM employs the maximum mean discrepancy loss to enhance concept discrimination within the concept space and facilitate accurate concept inference. An experimental evaluation on benchmark datasets (CUB-200-2011, TravelingBirds, AwA2, CheX-pert, and CelebA) demonstrates that EE-CBM achieve state-of-the-art performance in both concept accuracy and interpretability. This work positions the EE-CBM as a significant advancement in CBM researches, enabling them to effectively balance performance and interpretability for improved model transparency. Our code is available at https://anonymous.4open.science/r/EE-CBM-F48D.

## 1 Introduction

Model interpretation is increasingly important because of the opaque nature of deep learning models, particularly in critical image-based domains such as healthcare and autonomous driving. Concept bottleneck models (CBM) (Koh et al., 2020; Espinosa Zarlenga et al., 2022; Yuksekgonul et al., 2023; Chauhan et al., 2023; Kim et al., 2023; Sarkar et al., 2022; Havasi et al., 2022) have emerged as a solution to this challenge; their aim is to make the decision-making process of models transparent by simplifying it into understandable concepts. CBM researches infer the key concepts used in the prediction, and then predict the final label using only the inferred concepts, as shown in Fig. 1 (a). This approach significantly enhances the transparency of the model using concepts that humans can directly understand.

Early researches on CBM aimed to ensure model transparency, but this often resulted in accuracy that was lower than that of black-box models. To address this issue, recent CBM researches have seen a shift towards using large backbone networks and deep layers to achieve superior performance, contrary to their original purpose. This trend sacrifices the transparency of the model, focusing solely on improving the accuracy of final label predictions. Therefore, it is imperative to develop algorithms that can bridge the performance gap with black-box models while maintaining model transparency, in line with the original objectives of CBM researches. Model interpretability refers to the ability of a model to provide human-understandable explanations for its predictions, ensuring transparency in decision-making processes. In contrast, concept accuracy quantifies the correctness of the intermediate concept representations inferred by the model. While these two aspects are distinct, they are closely related. High concept accuracy enhances interpretability by ensuring that the concepts used in explanations align with the ground truth. To address the trade-off between accuracy and interpretability, concept embedding models (CEM) (Espinosa Zarlenga et al., 2022) has been proposed. CEM is a modified CBM network (Koh et al., 2020) that incorporate both positive and negative semantics, as shown in Fig. 1 (b). Coop-CBM (Sheth & Ebrahimi Kahou,

2023) enhanced CBM performance by employing an auxiliary loss to develop rich and expressive concept representations for downstream tasks. Energy-based CBM (ECBM) (Xu et al., 2024) utilized a collection of neural networks to establish the collective energy associated with candidate tuples comprising input, concept, and class. Through this unified framework, tasks such as prediction, concept refinement, and the assessment of conditional dependencies are expressed as conditional probabilities derived from the integration of diverse energy functions. Recent prominent CBM studies (Xu et al., 2024; Sheth & Ebrahimi Kahou, 2023; Sarkar et al., 2022), have adopted an approach to enhance model accuracy that incorporates $x \rightarrow c \rightarrow y$ and $x \rightarrow y$ structures to learn the relationship between final labels and concepts (Fig. 1 (c)). While these methods can improve label accuracy, often struggle to accurately infer concepts, a core goal of CBM research. This hinders model transparency. For example, while a CBM model may accurately diagnose pneumonia, it may fail to detect lung lesions, thereby undermining clinical trust.

In this study, we introduce a novel approach, the Energy Ensemble CBM (EE-CBM), which aims to enhance the balance between inference accuracy and interpretability in concept learning, as depicted in Fig. 1 (d). The proposed EE-CBM comprises two branch modules, concept extraction and concept probability. The concept extraction branch predicts concept values $\mathbf{C}$ through fully connected (FC) layers, similar to other CBM models. The concept probability branch generates probability $\mathbf{P}$ using an energy-based mechanism (LeCun et al., 2006), enhancing concept accuracy. This branch plays a pivotal role in determining the likelihood of each concept within the input data. It serves as a probability estimator, assigning probability to each concept and reflecting their relevance and contribution to the overall representation.

To ensure robust concept inference even in challenging scenarios such as noisy or wild images, an EE-CBM integrates samples generated through Markov chain Monte Carlo (MCMC) (Nijkamp et al., 2020a; 2019; 2020b; Han et al., 2017) methods. The concept value $\mathbf{C}$ and probability $\mathbf{P}$ of each branch are combined in the energy ensemble gate (EEG) to generate the final concept. The EEG alleviates potential information bottleneck issues in CBM researches. Furthermore, to promote accurate concept learning, an EE-CBM applies the maximum mean discrepancy (MMD) (Anderson et al., 1994; Gretton et al., 2012) as a loss to each concept embedding. This loss function fosters orthogonality between concept features, which brings similar concepts closer in latent space and creating distinct spaces for different concepts, resulting in clearer concept inference outcomes. In downstream tasks across various datasets, the EE-CBM effectively addresses balance the between conceptual understanding and label prediction.

The main contributions of the proposed EE-CBM are as follows:

- We introduce EE-CBM, a new architecture that aims to balance task accuracy and interpretability in concept learning and consists of two branches: the concept extraction branch and concept probability branch.
- The EEG combines the concept values and concept probabilities. This combination helps address the potential information bottleneck issues present in conventional CBMs.
- To ensure robust concept inference, especially for challenging data such as noisy or wild images, the EE-CBM integrates samples generated through MCMC methods. This potentially improves the ability of the model to handle complex data.
- The MMD loss function promotes the distinctiveness of concepts in latent space, bringing similar concepts closer together and separating distinct concepts.
- The proposed EE-CBM has demonstrated excellent performance and model interpretability through experiments conducted on various benchmark datasets such as CUB-200-2011 (Welinder et al., 2010), AwA2 (Xian et al., 2018), CheXpert (Irvin et al., 2019), and CelebA (Liu et al., 2015).

## 2 BACKGROUND

### 2.1 CONCEPT BOTTLENECK MODELS

CBM (Koh et al., 2020) is an interpretable method that can ensure the transparency of artificial intelligence models and classify images using human-friendly concepts that can be intuitively understood. Initial CBM studies (Koh et al., 2020; Espinosa Zarlenga et al., 2022) were constructed with

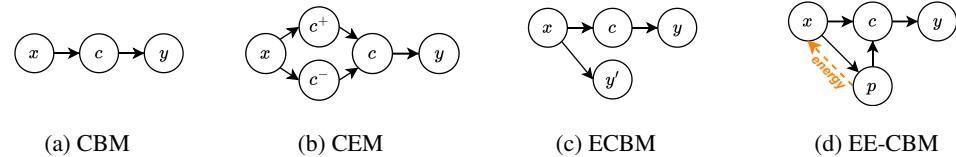

Figure 1: Structural differences in major CBM models. Our EE-CBM combines concept features and probabilities to address potential information bottleneck issues.

lightweight ResNet18 (He et al., 2016) or ResNet34 backbone networks and FC layers to emphasize model interpretability. However, recent studies (Xu et al., 2024; Sheth & Ebrahimi Kahou, 2023) have shown a trend towards the use of larger ResNet101 or Inception-v4 (Szegedy et al., 2017) networks, prioritizing classification performance over model interpretability.

The input data of CBM are composed of $\mathcal{D} = \{\mathcal{X}, \mathcal{C}, \mathcal{Y}\}$, which includes $N$ images $\mathcal{X} = \{x_1, x_2, \ldots, x_N\}$, concept labels $\mathcal{C} = \{\mathbf{C}_1^*, \mathbf{C}_2^*, \ldots, \mathbf{C}_N^*\}$, and class labels $\mathcal{Y} = \{y_1^*, y_2^*, \ldots, y_N^*\}$. Here, a single concept label $\mathbf{C}_n^* = \{c_1^*, c_2^*, \ldots, c_K^*\}$ consists of $K$ individual concept annotations $c_k^* \in \{0, 1\}$. CBM features a unified structure integrating two primary models. This structure follows the logic of $\mathcal{X} \to \mathcal{C} \to \mathcal{Y}$, comprising a concept encoder model, $h : \mathcal{X} \to \mathcal{C}$, which infers concepts from input images, and a model, $g : \mathcal{C} \to \mathcal{Y}$, which uses extracted concepts to infer the final class labels. Thus, the ultimate inference model is represented as $g(h(x))$. Each model $h$ and $g$ is trained to minimize the cross-entropy loss.

Concept labeling to reflect the detailed characteristics of objects may require expert knowledge, and understanding of the concepts could vary based on individual perspectives. Consequently, to address the problem of limited concept labeling, label-free approaches (Oikarinen et al., 2022; Wang et al., 2023; Yang et al., 2023; Shang et al., 2024) are being investigated. These methods are combined with large language models to generate concepts or enable the model to learn concepts by creating arbitrary concept embeddings. However, these approaches lead to an excessive increase in model parameters. Moreover, a semantic understanding of these models is difficult because heatmaps or other human-interpretable methods must be used to explain the concepts. Hence, initial CBM for image classification faces a trade-off between classification accuracy and model interpretability. To address this, several approaches have been proposed; however, existing methods (Sarkar et al., 2022; Sheth & Ebrahimi Kahou, 2023; Xu et al., 2024) tend to rely on large backbone networks to maintain accuracy, thereby neglecting the primary goal of CBM, which is model interpretability. In this study, we adopt energy-based models (EBM) to resolve this trade-off while maintaining the primary objectives of CBM—accuracy and model interpretability—regardless of the size of the backbone network.

## 2.2 ENERGY BASED MODELS

EBM is a model based on statistical physics principles such as the Boltzmann or Gibbs distributions (Ackley et al., 1985; Hinton et al., 2006; Salakhutdinov & Hinton, 2009). It is a powerful probabilistic model that can clearly model complex probability distributions. Unlike in conventional predictive models, in an EBM, lower energy corresponds to higher probability, and higher energy corresponds to lower probability. Energy in EBM can be expressed as follows.

$$p_\theta(x) = \frac{\exp(-E_\theta(x))}{Z(\theta)} \tag{1}$$

Here, probability $p_\theta$ is computed using energy function $E : \mathcal{X} \to e$ (i.e. $e \subseteq \mathbb{R}$) and partition function $Z(\theta)$. However, $Z(\theta)$ is an intractable term, and hence approximate sampling results are obtained using MCMC. For learning, EBM generates samples using Langevin dynamics sampling (Neal, 2011; Zhu & Mumford, 1998), which is an MCMC technique. Langevin dynamics sampling is obtained using the following equation.

$$\tilde{x}_t = \tilde{x}_{t-1} - \frac{\lambda}{2} \nabla_x E_\theta(\tilde{x}_{t-1}) + \epsilon_t, \quad \epsilon_t \sim \mathcal{N}(0, \lambda) \tag{2}$$

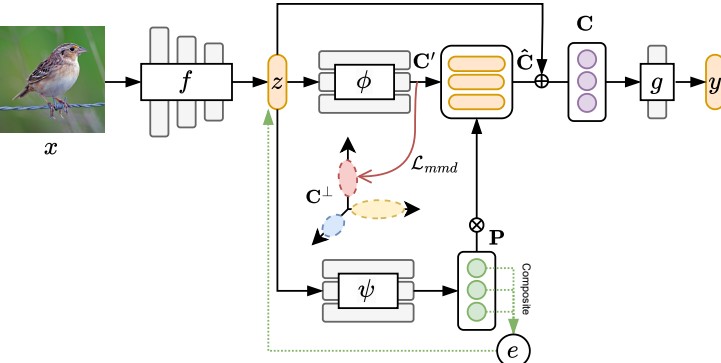

Figure 2: Overall architecture of the proposed EE-CBM. The input image is fed into a ResNet backbone network $f(x)$. The latent vector $z$ output by $f(x)$ is fed to both the concept extraction and concept probability branches. The outputs $\mathbf{C}'$ and $\mathbf{P}$ are combined into $\hat{\mathbf{C}}$ by the EEG. $\hat{\mathbf{C}}$ and $z$ are processed by the EEG to generate the final concept $\mathbf{C}$. The inferred final label $y$ is deduced via a single FC layer $g(\mathbf{C})$. The linearly composed energy $e$ is used for further training.

Here, $t$ denotes the number of steps, and $\epsilon_t$ represents the Gaussian noise at that step. Using these sampling techniques, we estimate the maximum likelihood to train an EBM. Given training images $\mathcal{X} = \{x_1, x_2, \ldots, x_N\} \sim p_{\mathcal{D}}(\mathcal{X})$ and samples $\tilde{\mathcal{X}} = \{\tilde{x}_1, \tilde{x}_2, \ldots, \tilde{x}_N\} \sim p_{\theta}(\tilde{\mathcal{X}})$ obtained through Langevin dynamic sampling, an EBM is trained using the following equation.

$$\nabla_\theta \mathcal{L}_e \approx \mathbb{E}_{x \sim p_{\mathcal{D}}}[\nabla_\theta E_\theta(x)] - \mathbb{E}_{\tilde{x} \sim p_\theta}[\nabla_\theta E_\theta(\tilde{x})] \tag{3}$$

We approximate Eq. (3) again so that it can be used for learning.

$$\nabla_\theta \mathcal{L}_e = \nabla_\theta \big[ \frac{1}{N} (\Sigma_{n=1}^N E_\theta(x_n) - \Sigma_{n=1}^N E_\theta(\tilde{x}_n)) \big] \tag{4}$$

The detailed derivation of this equation can be found in (LeCun et al., 2006), and a more comprehensive derivation is available in the Appendix A. EBM has demonstrated excellent results as a generative model (Gao et al., 2018; Guo et al., 2023; Zhao et al., 2017; Du et al., 2021; Pang et al., 2020; Du & Mordatch, 2019; Han et al., 2020), and it has also been used in classification tasks (Grathwohl et al., 2020; Kim & Ye, 2022; Yang & Ji, 2021; Guo et al., 2023). Therefore, by employing EBM in the concept encoder, we aim to achieve accurate concept inference and superior image classification performance.

## 3 EE-CBM

The proposed EE-CBM consists of the components depicted in Fig. 2. The key element is the concept encoder, which generates the final concept $\mathbf{C}$ and comprises two branches. The first branch, called the concept extraction branch, predicts concept value $\mathbf{C}'$ through FC layers as in conventional CBM models. The second branch, called the concept probability branch, employs an energy-based mechanism to determine the presence of each concept, producing probability $\mathbf{P}$ and enhancing concept accuracy. The resultant $\mathbf{C}'$ and $\mathbf{P}$ are combined into $\hat{\mathbf{C}}$. Subsequently, $\hat{\mathbf{C}}$ together with $z$, which is generated by the backbone network, is used to perform EEG to create the final concept $\mathbf{C}$, which avoids the concept information bottleneck. Finally, the inferred final label $y$ is deduced via a single FC layer. Additionally, we incorporate the MMD loss to ensure that the latent space of each concept remains similar and that each concept possesses orthogonal features.

## 3.1 CONCEPT ENCODER

**Concept encoder** $h(z) = \phi(z) \otimes \psi(z)$**.** The concept encoder includes two branches, $\phi(z)$ and $\psi(z)$, where $\phi : \mathbb{R}^d \to \mathbb{R}^u$ extracts concept features and $\psi : \mathbb{R}^d \to \mathbb{R}$ learns the probability of the concepts. Here, $d$ represents the number of output dimensions of the backbone network and $u$ represents the dimension of the concept. The concept features $\mathbf{C}' = \{c'_1, c'_2, \ldots, c'_K\}$ and concept probabilities $\mathbf{P} = \{p_1, p_2, \ldots, p_K\}$ produced by each branch are then combined into the final concept $\hat{c}$ using the EEG at the end.

**Concept feature extraction branch** $\phi(z)$**.** Branch $\phi(z)$ utilizes the feature vector $z \in \mathbb{R}^d$ extracted by backbone network $f(x)$ to generate concept features $c' \in \mathbb{R}^u$. As in CBM models, this branch is implemented through a single FC layer, and it can be represented by the following equation.

$$c'_k = \phi_i(z; \theta_{\phi_k}) = \text{FC}(f(x); \theta_{\phi_k}) \quad \text{s.t.} \quad c'_k \in \mathbb{R}^u, \quad k = 1, 2, ..., K \tag{5}$$

**Concept probability branch** $\psi(z)$**.** Branch $\psi(z)$ is responsible for calculating the probability for each concept. This is achieved using the EBM mechanism in which energy function $E_\theta(z)$ is implemented as a simple multi-layer perceptron. During the learning process of this branch, we utilize Langevin dynamics, a sampling technique in MCMC, to perform maximum likelihood learning. The adoption of MCMC in EBM is primarily motivated by the need for accurate and efficient sampling in complex concept representation spaces. This enables improved model performance and quantifiable uncertainty, leading to more reliable results. Branch $\psi(z)$ takes as input $z$, which has been extracted by backbone network $f(x)$. This branch can be represented by the following equation.

$$p_k = \psi_k(z; \theta_{\psi_k}) = E_{\theta_{\psi_k}}(z) \quad \text{s.t.} \quad p_k \in \mathbb{R}, \quad k = 1, 2, ..., K \tag{6}$$

To learn the energy, we employ maximum likelihood by replacing image $x$, which is the input to approximation Eq. (3), with latent vector $z$. Similarly, $\tilde{x}$ is also replaced with $\tilde{z}$, where $\tilde{z}$ is the vector sampled by applying MCMC to latent vector $z$ using Eq. (2). Using the modified approximation Eq. (3), we can save computational and memory costs required for training by using latent vector $z$ instead of images. Finally, in the loss function $\mathcal{L}_e$ (Eq. (4)), $x_i$ is replaced by $z$, and $\tilde{x}$ is replaced with by $\tilde{z}$ to learn the energy. At this point, the generated energy is linearly composed for use in energy learning, and the equation is as follows.

$$\nabla_\theta \mathcal{L}_e = \nabla_\theta \Big[ \frac{1}{N} (\Sigma_{n=1}^N E_{\theta_\psi}(z_n) - \Sigma_{n=1}^N E_{\theta_\psi}(\tilde{z}_n)) \Big] \tag{7}$$

The generated energy is used to compute the probability $p$ of the concept. Concept probability $p$ represents the probability that concept $c$ exists given input data $z$. The structure proposed in this study employs the concept probability branch, which consists of an EBM mechanism based on MCMC techniques. This offers a more practical approach to learning the concept probabilities that exist in images in the wild than existing CBM models.

**EEG.** In conventional EBM methods, a concept information bottleneck can occur in which only specific concepts are selectively learned during the model learning process. Hence, it becomes challenging to discern the deep associations among concepts, leading to constraints on the representation. To address this issue, we propose the EEG, which enables the overall context $z$ to be flexibly combined with the concepts. In other words, the EEG effectively combines $z$ with the concept features $\hat{\mathbf{C}}$ and concept probabilities $\mathbf{P}$ generated in the two branches to produce the final concept $\mathbf{C}$. In the EEG, hidden connections between concepts are learned and their representation is improved. The equation for EEG learning is as follows.

$$c_k = \underbrace{(\sigma(p_k) \cdot W_p) \otimes (c'_k \cdot W_c)}_{\hat{c}_k} + z \cdot W_z, \quad \mathbf{C} = \{c_1, c_2, ..., c_K\} \tag{8}$$

Here, $W_c \in \mathbb{R}^{u \times u}$, $W_p \in \mathbb{R}$ and $W_z \in \mathbb{R}^{d \times u}$ are trainable weight parameters, $\sigma(p)$ denotes the sigmoid function, which is computed using $\frac{1}{1+\exp(-p)}$, and $\otimes$ represents the multiplication operation.

As Eq. (8) reveals, the EEG flexibly integrates $z$ to address the concept information bottleneck problem, which leads to a limited representation due to compacted information. The learned final concept $\mathbf{C}$ is passed through a single FC layer to generate the final prediction $y$. Throughout the entire model training process, concept $\mathbf{C}$ improves prediction accuracy through binary cross-entropy loss $\mathcal{L}_c$. Additionally, $y$ is used to train the precise multi-class classification using cross-entropy loss $\mathcal{L}_y$. The overall loss required for training is as follows.

$$\mathcal{L}_{eeg} = \lambda_c \mathcal{L}_c + \lambda_y \mathcal{L}_y + \lambda_e \mathcal{L}_e \tag{9}$$

where, $\lambda_c$, $\lambda_y$, and $\lambda_e$ are the hyperparameters for each loss, which enable us to adjust the importance of each loss function to optimize model performance. More detail explanation of $\lambda_c$, $\lambda_y$, and $\lambda_e$ can be found in the Appendix B. The pseudocode of the entire algorithm is presented in Algorithm 1.

### 3.2 CONCEPT MMD LOSS

In this paper, we employ the total loss function $\mathcal{L}_{eeg}$ to train the energy probability model, final concepts, and final predictions. However, relying on this loss function alone may prove insufficient for effective concept learning.

$$\mathcal{L}_{mmd} = \frac{1}{K} \sum_{k=1}^{K} \| \mu(c_k^{\perp} | c_k^*) - \mu(c_k' | c_k^*) \|_2^2 \tag{10}$$

Here, $c_k^{\perp} \in \mathbf{C}^{\perp}$ denotes the orthogonal latent vector, i.e., variational concept conditional marginal, and $c_k' \in \mathbf{C}'$ represents the predicted concept feature. $\mu$ is a kind of mapping function (e.g. batch-wise average). By introducing the $\mathcal{L}_{mmd}$ loss, we encourage the concept learning process to train feature spaces where each concept is both similar to itself and distinctly separable from other concepts. Consequently, the $\mathcal{L}_{total}$ of the proposed model is modified as follows.

$$\mathcal{L}_{total} = \mathcal{L}_{eeg} + \lambda_{mmd} \mathcal{L}_{mmd} \tag{11}$$

---

**Algorithm 1** EE-CBM algorithm

---

**Input** : input image $x$, # of concepts $K$

$z = f(x)$ // extract feature vector $z$ from backbone $f(\cdot)$
$\mathbf{C} = \emptyset$    // init concept set
**for** $k \in \{1, 2, ..., K\}$ **do**
    $c'_k = \phi_k(z)$    // concept feature extraction branch
    $p_k = \psi_k(z)$    // concept probability branch
    $c = (\sigma(p_k) \cdot W_p) \otimes (c'_k \cdot W_c) \oplus z \cdot W_z$
    // operate energy ensemble gate

    $\mathbf{C} = \mathbf{C} \cup c$

$y = g(\mathbf{C})$ // predict class label $y$ from $g(\cdot)$

**Output** : $(\mathbf{C}, y)$

---

## 4 EXPERIMENTS

To evaluate the performance of our proposed model, we conducted experiments using four datasets: **CUB-200-2011** (Welinder et al., 2010), **TravelingBirds** (Koh et al., 2020), **AwA2** (Xian et al., 2018), **CheXpert** (Irvin et al., 2019), and **CelebA** (Liu et al., 2015). The **CUB-200-2011** dataset consists of 11,788 images belonging to 200 categories of birds. It is divided into a training set of 5,994 images and a test set of 5,794 images. Each image is annotated with one category label and 312 attributes (concepts). We followed the approaches of CBM and CEM, utilizing 112 attributes as concepts and using the same data partitioning. The **TravelingBirds**, a segmented bird image dataset derived from CUB, offers a diverse range of background conditions. This dataset is categorized into CUB Random, CUB Fixed, and CUB Black, and it is particularly useful for evaluating a model's ability to encode object-centric concepts while minimizing the impact of background variations. The **AwA2** dataset comprises 37,322 images of 50 animal categories, with 85 attributes. The **CheXpert** dataset includes 224,316 chest radiographs from 65,240 patients, with two category labels and 13 attributes. The CheXpert dataset provides concept uncertainty labels, which were incorporated during training to address ambiguous concepts effectively. The **CUB-200-2011** and **AwA2** datasets are widely used benchmarks for models using attributes, as they contain a relatively large number of

concepts. The **CheXpert** dataset, by contrast, includes two attributes and incorporates the uncertainty of the concepts. The **CelebA** dataset contains 202,599 face images of 10,177 celebrities, along with 40 attributes. Please see Appendix E for detail information on the five datasets.

Table 1: Comparison of the accuracy results of the comparison models on three dataset. The results of experiments conducted using five different seeds are reported. Additionally, the experiments are performed using two different sizes of the backbone network. The best performance is in **bold**, and the second-best performance is underlined. The symbols † and ‡ indicate ResNet34 and ResNet101 backbone, respectively. Comparison methods were trained using the exact strategies and configurations recommended by the original papers. (The performance results of CelebA are presented in Appendix C.)

| Methods | CUB | | CheXpert | | AWA2 | |
|---|---|---|---|---|---|---|
| | Concept (%) | Task (%) | Concept (%) | Task (%) | Concept (%) | Task (%) |
| Bool-CBM† | 96.229 (±0.031) | 72.512 (±0.466) | 84.428 (±1.121) | 83.682 (± 0.000) | 99.001 (±0.188) | 94.868 (±1.047) |
| Fuzzy-CBM† | 95.882 (±0.105) | 74.228 (±0.606) | 83.740 (±0.718) | 81.916 (±1.448) | 98.999 (±0.167) | 95.088 (±1.004) |
| CEM† | 96.159 (±0.156) | 79.029 (±0.518) | 84.315 (±1.247) | 82.125 (±2.604) | 99.048 (± 0.036) | 95.745 (±0.293) |
| Prob-CBM† | 95.596 (±0.061) | 76.265 (±0.145) | 86.692 (± 0.123) | 83.652 (±0.083) | 98.283 (±0.065) | 92.484 (±0.315) |
| ECBM† | 96.536 (± 0.091) | 77.148 (±0.695) | 84.792 (±0.842) | 83.682 (± 0.000) | 98.908 (±0.037) | 94.555 (±0.120) |
| Coop-CBM† | 89.892 (±0.649) | 79.154 (± 0.734) | 84.435 (±0.201) | 82.993 (±1.244) | 98.875 (±0.107) | 95.927 (± 0.153) |
| Ours (EE-CBM†) | **96.554 (± 0.057)** | **80.417 (± 0.291)** | **86.703 (± 0.236)** | **87.145 (± 0.145)** | **99.063 (± 0.005)** | **96.218 (± 0.435)** |
| Bool-CBM‡ | 96.602 (± 0.310) | 75.784 (± 0.204) | 84.703 (± 1.222) | 83.682 (± 0.000) | 99.227 (± 0.100) | 95.547 (± 0.697) |
| Fuzzy-CBM‡ | 96.442 (± 0.104) | 78.523 (± 1.133) | 85.179 (± 0.743) | 84.584 (± 0.811) | 99.102 (± 0.054) | 95.757 (± 0.242) |
| CEM‡ | 96.585 (± 0.102) | 80.755 (± 0.287) | 84.476 (± 1.416) | 84.530 (± 0.597) | 99.201 (± 0.030) | 96.235 (± 0.204) |
| Prob-CBM‡ | 96.614 (± 0.137) | 77.372 (± 0.931) | 86.722 (± 0.151) | 83.682 (±0.083) | 98.414 (± 0.044) | 92.922 (± 0.214) |
| ECBM‡ | 96.661 (± 0.262) | 79.426 (± 0.241) | 85.256 (± 0.351) | 83.682 (± 0.000) | 99.078 (± 0.040) | 95.431 (± 0.173) |
| Coop-CBM‡ | 91.340 (± 1.419) | 81.106 (± 0.695) | 84.265 (± 0.367) | 84.131 (± 2.044) | 99.048 (± 0.107) | 95.927 (± 0.153) |
| Ours (EE-CBM‡) | **96.696 (± 0.037)** | **81.141 (± 0.139)** | **86.733 (± 1.532)** | **87.379 (± 1.023)** | **99.230 (± 0.113)** | **96.440 (± 0.525)** |

As mentioned earlier, to demonstrate the consistent performance of CBM regardless of the size of the backbone network, we opted not only for ResNet101 used in existing CBM methods but also additionally selected ResNet34, a smaller backbone network. The images in the datasets were resized to 299×299, and the SGD optimizer was employed. Detailed hyperparameter settings are provided in the Appendix B.

## 4.1 QUANTITATIVE EXPERIMENTS

**Trade-off between task accuracy and interpretability.** To evaluate the effectiveness of the proposed model, we present the results of our experiments on task (classification) accuracy, concept interpretability. Our experimental protocol follows the same procedure as CEM to ensure a fair comparison between different methods. Table 1 presents the results of the trade-off experiment for downstream tasks on three datasets. The experiments are conducted by varying the size of the backbone network between ResNet34† and ResNet101‡. Across all datasets, the EE-CBM consistently achieves significantly higher performance in both metrics than the other models. In particular, because it uses concept probabilities, the EE-CBM demonstrates significantly higher performance than the other models on the CheXpert dataset, which includes the uncertainty of the concepts.

First, we examine the performance of the methods that use ResNet34 as the backbone. In CEM, both positive and negative concepts are utilized, and these concepts are represented as vectors rather than scalars. Due to its ability to capture rich information about concepts, CEM achieved good performance in terms of task accuracy and interpretability on AwA2 dataset. In contrast, EE-CBM, despite using scalar concepts, resolved the concept information bottleneck problem, thereby enhancing both accuracy and interpretability and achieving the highest performance on all datasets. Despite employing additional $x \rightarrow y$ loss functions, Coop-CBM and ECBM demonstrated lower performance compared to EE-CBM. These results are consistently observed in experiments conducted with the backbone changed to ResNet101. This confirms that the proposed EE-CBM outperforms existing models across both metrics, demonstrating consistent performance regardless of the backbone size. Ultimately, the ability of EE-CBM to accurately capture and clearly explain concepts allows it to overcome the trade-off between task accuracy and interpretability that has hindered previous methods.

Table 2: Quantitative comparison of task accuracy (%) on background-shifting datasets (Traveling-Birds) using various concept bottleneck design models. The best performance is in **bold**, and the second-best performance is underlined.

| Methods | CUB Black | | CUB Random | |
| --- | --- | --- | --- | --- |
| | Concept (%) | Task (%) | Concept (%) | Task (%) |
| Bool-CBM | 93.032 (±0.662) | 55.346 (±3.282) | 92.855 (±0.139) | 55.357 (± 0.882) |
| Fuzzy-CBM | 93.028 (±0.383) | 59.561 (±2.011) | 92.519 (±0.107) | 58.460 (±0.356) |
| CEM | 92.903 (±0.548) | 60.973 (±2.671) | 92.666 (±0.166) | 62.388 (±1.257) |
| Prob-CBM | 93.280 (±0.177) | 65.364 (±1.058) | 91.942 (± 0.201) | 59.506 (±0.646) |
| ECBM | 94.124 (± 0.286) | 60.472 (±1.346) | 93.187 (±0.283) | 56.320 (± 1.207) |
| Coop-CBM | 88.301 (±0.665) | 63.994 (± 1.270) | 87.840 (±0.540) | 61.663 (±1.286) |
| Ours (EE-CBM) | **94.568 (± 0.001)** | **69.825 (± 0.012)** | **93.744 (± 0.002)** | **66.960 (± 0.012)** |

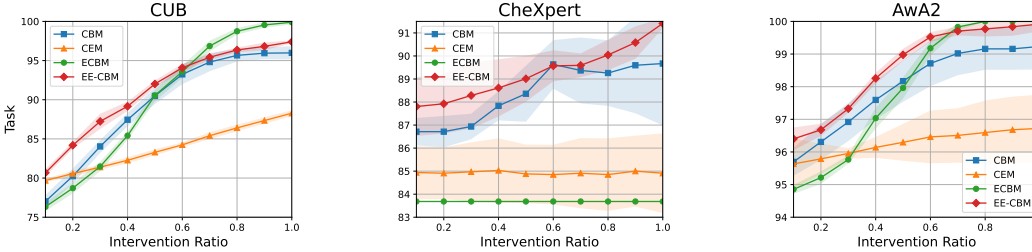

(a) CUB      (b) CUB Black      (c) CUB Random

Figure 3: Representative samples from the CUB and TravelingBirds datasets with background manipulations for evaluating model generalization. (a) CUB, (b) CUB Black, (c) CUB Random.

**Background shifting.** Figure 3 presents sample images from the CUB Black and CUB Random subsets of the TravelingBirds dataset. These images demonstrate the effects of shifting backgrounds to a uniform black color or to random real-world scenes. Background shifting was employed to simulate more challenging scenarios, such as introducing changes in background appearance, altering the spatial distribution of background elements, and varying the level of background complexity. By subjecting the model to these controlled shifting, we were able to assess its resilience to a wider range of background shifts. To comprehensively evaluate the robustness of our proposed model against background shifts, we conducted quantitative experiments on the TravelingBirds dataset, a variant of the CUB dataset. Our experimental setup involved training the EE-CBM model based on ResNet34 using the CUB dataset and testing it on various background-shifting subsets of TravelingBirds, including CUB Black and CUB Random. As shown in Table 2, the proposed EE-CBM consistently outperformed the baseline models across CUB Black and CUB Random background-shifting datasets. These results highlight the model's remarkable robustness and generalization capabilities. Unlike other methods that may be influenced by background information, our proposed method exhibits a greater ability to focus on the conceptual content of the image, irrespective of the background. This enhanced conceptual focus allows the model to deliver more reliable interpretations, consistently achieving better performance even under varying and dynamic background conditions. (For additional details, please refer to Appendix H.)

Figure 4: Task accuracy according to type of concept intervention.

**Concept intervention.** Figure 4 presents the performance when concept intervention techniques are used on each dataset. Intervention experiments involve artificially modifying predicted concepts to assess their causal influence on model output, thereby revealing the underlying relationships between concepts. These experiments highlight the model's transparency and aid in understanding the rationale behind specific decisions. To ensure fairness across all compared methods, we did not

apply the random concept intervention strategy during training. Instead, we conducted intervention experiments by randomly selecting a concept intervention ratio between 0.1 and 1.0 within the total number of concepts in the given dataset (see Appendix I). In the CUB dataset, the proposed EE-CBM based on ResNet34 demonstrates highest performance when the intervention ratio is low, but slightly lower performance compared to ECBM after 0.6. However, EE-CBM consistently outperforms other methods regardless of ratio changes in the CheXpert dataset. The ECBM, which shows high performance in CUB, consistently exhibits lower performance in CheXpert, indicating sensitivity to data types in terms of concept intervention. In experiments on the AwA2 dataset, EE-CBM also exhibits the best performance. Through experiments on the three datasets, we can confirm that EE-CBM, based on an energy-based probability model, generates concepts robust to uncertainty. In addition, EE-CBM consistently demonstrates excellent intervention performance even when the dataset type changes and the number of specified concepts varies across datasets. Based on this, it can be interpreted that the EE-CBM correctly understands intuitive concepts understood by humans and uses them as the basis for class inference. As for concept accuracy, the EE-CBM demonstrates high task accuracy across all datasets. This can also be interpreted as a result of its outstanding understanding of concepts.

**Concept importance.** This experiment aimed to verify whether the proposed EE-CBM accurately discerns the presence of concepts in input images. In particular, we focused on confirming whether the EE-CBM provides precise concept probabilities through the concept probability branch. The experiment involved calculating the concept probabilities for various images using the EE-CBM based on ResNet34 and conducting a comparative analysis with the actual presence of concepts.

Figure 5 shows examples demonstrating these experiments. In Fig. 5, the EE-CBM maintains high accuracy even on challenging images such as images with complex backgrounds or partially occluded objects. Therefore, it can be concluded that the EE-CBM provides precise concept probabilities because of its concept probability branch. This ultimately serves as more evidence that the EE-CBM comprehends concepts accurately and performs precise label predictions based on this comprehension.

**MMD loss.** To verify whether MMD loss indeed improves model performance, we conducted an ablation study. As shown in Table 3, when MMD loss was not utilized ($\lambda_{mmd} = 0$), there was a 1.025% decrease in concept accuracy. This indicates the importance of MMD loss in concept learning. In this scenario, the decrease in concept accuracy also led to a 0.774% decrease in task accuracy. From these results, we can infer that a thorough understanding of concepts is essential for enhancing task accuracy.

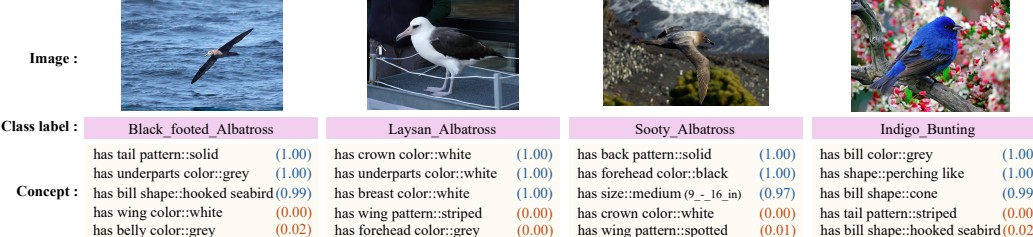

Figure 5: Visualization of the concept probabilities for each image sample. The top three concepts represent the results of accurately inferring concepts that are confidently present in the image, while the bottom two represent confidently inferring the absence of concepts not present in the image.

## 4.2 ABLATION STUDIES

This section presents an evaluation of the effectiveness of each module in the proposed EE-CBM based on ResNet34. This ablation studies were conducted using the CUB-200-2011 dataset.

Table 3: Ablation study performance comparison results for the Concept probability branch ($\psi(z)$), EEG ($\lambda_e$) and MMD loss ($\lambda_{mmd}$) in the proposed EE-CBM based on ResNet34.

| Variants | Concept (%) | Task (%) |
|---|---|---|
| EE-CBM | **96.554 ($\pm$ 0.057)** | **80.417 ($\pm$ 0.291)** |
| *w/o* $\psi(z)$ | 95.253 ($\pm$0.106) | 77.789 ($\pm$0.445) |
| $\lambda_{mmd} = 0$ | 95.124 ($\pm$0.309) | 78.408 ($\pm$0.388) |
| $\lambda_e = 0$ | 95.080 ($\pm$0.380) | 77.365 ($\pm$0.523) |

**EEG.** We use the EEG module to flexibly integrate the outputs of the concept probability and concept feature extraction branches, thereby addressing the concept information bottleneck while enhancing

model performance. To evaluate the extent to which the concept probability branch, a key component of the EE-CBM, influences the EEG, we conducted experiments comparing the performance with and without the core energy learning of the EEG module ($\lambda_e = 0$). As evident in Table 3, using the EEG module resulted in a 1.817% improvement in performance. In particular, a significant enhancement in concept accuracy was observed. This indicates that the EEG module alleviated the concept information bottleneck, enabling the model to better comprehend concepts. Furthermore, as concept accuracy increased, task accuracy also improved. This demonstrates that with a better understanding of concepts, the model can infer task accuracy more accurately.

As indicated in Table 3, the removal of the proposed concept probability branch led to a general decline in performance. Also the difference between the performance without the branch and that with only the energy loss $\mathcal{L}_e$ removed is relatively small, suggesting that the energy loss has a more limited impact on performance. Nevertheless, the consistent performance improvements observed when the branch is included indicate that it contributes significantly to the overall model performance. This finding suggests that while the concept probability branch may not be the sole determinant of performance, it plays a supportive role in the learning process by facilitating the flow of information between the concept probabilities and the concept values. The concept probability branch, updated through the energy loss $\mathcal{L}_e$, plays a crucial role in effectively communicating clear concept probabilities $\mathbf{P}$ to the concept values $\mathbf{C}$. Without this branch, the learning process becomes less efficient, resulting in suboptimal performance.

**Orthogonal concept latent space.** To determine whether MMD loss effectively learns a latent space in which similar concepts are close to each other and different concepts are positioned farther apart, we visualized the latent space of five random concepts. Figure 6 (a) depicts the distribution of the concepts using MMD loss, while Fig. 6 (b) illustrates the distribution obtained without using MMD loss. When MMD loss is employed, the characteristics of each concept are clearly separated. In contrast, when MMD loss is not used, the concepts appear to be mixed. This demonstrates that MMD loss aids in effectively classifying concepts.

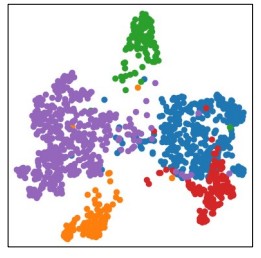 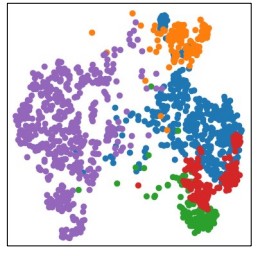

(a) EE-CBM *w/* MMD loss      (b) EE-CBM *w/o* MMD loss

Figure 6: Visualization of the t-SNE (Van der Maaten & Hinton, 2008) latent spaces for five random concepts (a) using the MMD loss and (b) excluding the MMD loss.

## 5 DISCUSSION AND LIMITATION

This work introduced the EE-CBM, which effectively addresses the trade-off between conceptual understanding and label prediction in downstream tasks. The core concept of the EE-CBM lies in its leveraging of energy ensembles and concept probability to tackle the concept information bottleneck regradless of the backbone size. This approach enables the model to achieve a deeper grasp of concepts. Furthermore, because it incorporates the MMD loss, the EE-CBM facilitates the formation of a latent space in which similar concepts are positioned close together, whereas distinct concepts are separated by a larger distance. The experimental results establish the EE-CBM as a promising CBM because it achieves high concept accuracy and interpretability results across all datasets.

It is encouraging that EE-CBM demonstrates consistent performance not only in model interpretability but also in task accuracy compared to existing complex black-box models. However, constructing datasets that include concepts entails significant costs, and because of the limited concept resources within datasets, there could be instances where the model fails to learn the concepts actually necessary for training. To address this, future research must focus not only on improving task accuracy but also on exploring concept-free models. Furthermore, although the proposed model benefits from using MCMC for energy learning, which allows it to extract concepts well from images in the wild, it suffers from the drawback of multiple iterations. Therefore, energy-efficient learning algorithms should be developed in future to overcome the limitations of the EE-CBM and enhance its performance.

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

## A    APPENDIX: DERIVATION OF ENERGY-BASED METHOD

As mentioned in Section 2.2 of this paper, here we discuss the approximation methods for the EBM equation.

$$p(x) = \frac{1}{Z(\theta)}\exp(f_\theta(x)) \tag{12}$$

$Z_\theta$ can be expressed as follows.

$$Z(\theta) = \int \exp(f_\theta(x))dx \tag{13}$$

Under the Maximum Likelihood Estimation (MLE) condition, the loss function of the EBM can be defined as follows.

$$\mathcal{L}_e = \frac{1}{n}\sum_{i=n}^{N}\log(p_\theta(x)) \tag{14}$$

For training, it is necessary to obtain the derivative $\mathcal{L}_e'$ (i.e., $\nabla_\theta \mathcal{L}_e$) of the loss function. The derivative $\mathcal{L}_e'$ of the loss function can be calculated as follows.

$$\nabla_\theta \mathcal{L}_e = \frac{1}{n}\sum_{i=n}^{N}\nabla_\theta\log(p_\theta(x)) \tag{15}$$

$$= \frac{1}{n}\sum_{i=n}^{N}\nabla_\theta\log(\frac{1}{Z}\exp(f(x))) \quad \because \quad p_\theta(x) = \frac{1}{Z}\exp(f(x)) \tag{16}$$

$$= \mathbb{E}_x\nabla_\theta(-\log(Z) + f(x)) \tag{17}$$

$$= \mathbb{E}_x\nabla_\theta f(x) - \mathbb{E}_x\nabla_\theta\log(Z) \tag{18}$$

$$= \mathbb{E}_x\nabla_\theta f(x) - \nabla_\theta\log(Z) \tag{19}$$

Although we can calculate the derivative as shown in Eq. (19), the $\log(Z)$ term cannot be directly computed. Therefore, we approximate the $\log(Z)$ term with a computable expression.

$$\nabla_\theta\log(Z) = \frac{1}{Z(\theta)}\nabla_\theta Z(\theta) \tag{20}$$

$$= \frac{1}{Z(\theta)}\nabla_\theta\int\exp(f_\theta(x))dx \quad \because \quad Z(\theta) = \int\exp(f(x))dx \tag{21}$$

$$= \frac{1}{Z(\theta)}\int\nabla_\theta\exp(f_\theta(x))dx \quad \because \text{swap } \nabla_\theta \text{ and } \int \tag{22}$$

$$= \frac{1}{Z(\theta)}\int\exp(f_\theta(x))\nabla_\theta f_\theta(x)dx \tag{23}$$

$$= \int\frac{1}{Z(\theta)}\exp(f_\theta(x))\nabla_\theta f_\theta(x)dx \tag{24}$$

$$= \int p_\theta(x)\nabla_\theta f_\theta(x)dx \quad \because \quad p(x) = \frac{1}{Z(\theta)}\exp(f_\theta(x)) \tag{25}$$

$$= \mathbb{E}_{p_\theta(x)}[\nabla_\theta f_\theta(x)] \tag{26}$$

Finally, if Eq. (26) is substituted into Eq. (19) and developed, the following final equation can be obtained. (At this time, $f(x)$ is an energy model, so it is possible to express it as $E(X)$.)

By substituting equation Eq. (26) into Eq. (19), we can obtain the result of Eq. (27). Here, $f(x)$ can be expressed as the energy model $E(x)$ (Eq. (28)), and discretizing the expectation yields the final result as shown in equation (Eq. (29)).

$$\therefore \quad \nabla_\theta \mathcal{L}_e = \mathbb{E}_x[\nabla_\theta f(x)] - \mathbb{E}_{p_\theta(x)}[\nabla_\theta f_\theta(x)] \tag{27}$$

$$\nabla_\theta \mathcal{L}_e \approx \mathbb{E}_x[\nabla_\theta E_\theta(x)] - \mathbb{E}_{\tilde{x}}[\nabla_\theta E_\theta(\tilde{x})] \tag{28}$$

$$\mathbb{E}_x[\nabla_\theta E_\theta(x)] - \mathbb{E}_{\tilde{x}}[\nabla_\theta E_\theta(\tilde{x})] = \nabla_\theta[\frac{1}{N}(\Sigma_{n=1}^N E_\theta(x_n) - \Sigma_{n=1}^N E_\theta(\tilde{x}_n))] \tag{29}$$

## B  APPENDIX: HYPERPARAMETERS

We used slightly different hyperparameters for each dataset as shown in Table 4. The training hyperparameter values presented in Table 4 were determined by setting multiple candidate values, training all possible combinations, and selecting the combination that yielded the best performance as the final hyperparameters.

Table 4: Hyperparameters used for training

| Hyperparameter | Dataset | | | |
| --- | --- | --- | --- | --- |
| | CUB | CelebA | AwA2 | CheXpert |
| Learning rate | | | 0.001 | |
| $\lambda_c$ | 5.5 | 7.5 | 7.5 | 7.5 |
| $\lambda_y$ | | | 3 | |
| $\lambda_e$ | | | 0.1 | |
| $\lambda_{mmd}$ | | | 0.1 | |
| dim $u$ | | | 16 | |
| Batch size | | | 32 | |
| Epoch | | | 300 | |
| Optimizer | | | SGD | |
| Weight decay | | | 4.0e-5 | |
| Momentum | | | 0.9 | |
| Input resolution | | | 299 | |

# C Appendix: CelebA performance experiment

Table 5: Comparison of the accuracy results of the comparison models on CelebA dataset.

| Methods | CelebA | |
|---|---|---|
| | Concept Acc. (%) | Task Acc. (%) |
| Bool-CBM | 90.329 (±0.164) | 33.915 (±0.884) |
| Fuzzy-CBM | 90.269 (±0.211) | 33.696 (±2.103) |
| CEM | 90.237 (±0.306) | **42.617 (±1.412)** |
| Prob-CBM | 89.271 (±0.238) | 34.472 (±0.839) |
| ECBM | 90.006 (±0.986) | 34.975 (±2.111) |
| Coop-CBM | 90.533 (±0.142) | 42.392 (±1.354) |
| Ours (EE-CBM) | **90.699 (±0.656)** | 35.203 (±0.766) |

We present the results of experiments on the CelebA dataset. The CelebA dataset is a large-scale dataset with labeled facial images and 40 attributes. In this experiment, we extracted six key facial attributes from the CelebA dataset. These limitations can lead to performance degradation. In fact, we found that models that use concept scalars instead of concept vectors typically achieve a low task accuracy of 33-34%. In contrast, the proposed model achieves a task accuracy of 35.203%, which is the highest among concept scalar models and represents an approximately 1% improvement over previous models. This shows that the proposed model exceeds the performance limitations of concept scalar models.

# D Appendix: Model Complexity

In this section, we present a comparative analysis of the computational complexity of the proposed EE-CBM and the baseline models. While EE-CBM introduces a modest increase in the number of parameters owing to the Markov Chain Monte Carlo (MCMC) sampling for the energy function, it demonstrates a lower computational complexity in terms of floating-point operations per second (FLOPs) compared to Prob-CBM. Furthermore, our experimental results reveal that EE-CBM achieves the lowest latency when evaluated under identical system conditions. This finding suggests that EE-CBM offers a compelling balance between model performance and computational efficiency.

Table 6: Comparison of the complexity of the comparison models.

| Methods | FLOPs (G) | Latency (ms) |
|---|---|---|
| Fuzzy-CBM | 6.85 | 4.86 |
| CEM | 6.85 | 8.74 |
| Prob-CBM | 7.38 | 49.38 |
| ECBM | 6.84 | 23.17 |
| Coop-CBM | 6.84 | 5.86 |
| Ours (EE-CBM) | **7.36** | **5.86** |

# E  APPENDIX: EXPERIMENTAL SETUP AND ENVIRONMENTAL DETAILS

## CODE, MODELS, AND LICENSES

Our implementation was carried out in Python 3.9 using various open-source libraries, including PyTorch 1.12.1 (BSD license), torchvision 0.13.1 (BSD license), Scikit-learn 1.2.1 (BSD license), and OpenCV 4.7.0 (BSD license). For visualizations, we used Matplotlib 1.3.0 (BSD license). To ensure the reproducibility of our experiments, we have made all relevant code publicly available in a repository under the MIT license.

## RESOURCES

All of the experiments were conducted on a private machine equipped with two Intel(R) Xeon(R) CPUs, that is, a Gold 6230R CPU @ 2.10 GHz; 128 GB RAM, and an NVIDIA RTX 3090 GPU.

## DATASET DESCRIPTION

**CUB** (Welinder et al., 2010) dataset contains images of 200 bird species. The dataset consists of a total of 11,788 images, with 5,994 training images, and 5,794 testing images. Each image is labeled with 112 attributes, representing various characteristics of each bird species.
**CelebA** (Liu et al., 2015) dataset is a facial image dataset consisting of a total of 202,599 images from 10,177 celebrities. This dataset includes images taken in various situations, so each image has different facial expressions, lighting, clothing, and so on. Each facial image in the dataset is labeled with 40 attributes, representing various characteristics such as gender, eyeglasses, and hats. However, some attributes have uncertain labeling. Therefore, as with CEM, the eight attributes with the highest normal distributions were selected, and two of these attributes were trained without labels. As a result, the total number of classes is about 240, and only six attributes were optimized using ground-truth labels.
**AwA2** (Xian et al., 2018) dataset is designed for attribute-based and zero-shot learning tasks. It contains 37,322 images across 50 animal classes, each annotated with 85 attribute labels.
**CheXpert** (Irvin et al., 2019) dataset is a large-scale dataset for chest radiograph interpretation, designed to facilitate research in medical image analysis and automated diagnosis. It contains 224,316 chest radiographs from 65,240 patients, labeled for 14 common chest conditions such as atelectasis, cardiomegaly, and pleural effusion. The dataset includes uncertainty labels to account for ambiguous cases, with conditions annotated as positive, negative, or uncertain. CheXpert also provides a standardized validation set with expert-annotated labels for model evaluation. The dataset is split into training, validation, and test sets, enabling robust assessment of model performance.
**TravelingBirds** (Koh et al., 2020) dataset is a synthetic dataset derived from the CUB dataset, created to assess the robustness of models under real-world conditions. By replacing the original backgrounds of CUB images with a variety of diverse scenes, the TravelingBirds dataset introduces a level of uncertainty that mimics the challenges faced by models deployed in real-world applications. This dataset is particularly useful for evaluating the generalization ability of models and their ability to handle variations in background complexity.

# F  APPENDIX: IMPACT OF CONCEPT REDUCTION ON TASK AND CONCEPT ACCURACY

The results presented in Table 7 demonstrate the impact of reducing the number of concepts on both task accuracy and concept accuracy for the CUB dataset. The experiments compare Fuzzy-CBM, CEM, and EE-CBM when trained and evaluated with 100 concepts versus 50 randomly selected concepts.

As shown in the table, our proposed model (EE-CBM) exhibits the smallest decline in both task accuracy and concept accuracy when the number of concepts is reduced. This underscores the

Table 7: Impact of Concept Reduction on Task and Concept Accuracy for the CUB Dataset. Performance comparison of Fuzzy-CBM, CEM, and EE-CBM when trained and evaluated with 100 concepts versus 50 randomly selected concepts. Results include concept accuracy (Concept Acc.) and task accuracy (Task Acc.) with mean and standard deviation over multiple runs.

| Methods | 50 concepts | | 100 concepts | |
|---|---|---|---|---|
| | Concept Acc. (%) | Task Acc. (%) | Concept Acc. (%) | Task Acc. (%) |
| Fuzzy-CBM | 96.15 ($\pm$0.02) | 66.90 ($\pm$0.18) | 95.72 ($\pm$0.02) | 73.26 ($\pm$0.56) |
| CEM | 96.09 ($\pm$0.01) | 77.36 ($\pm$0.19) | 95.85 ($\pm$0.17) | 78.89 ($\pm$0.14) |
| Ours (EE-CBM) | **96.99 ($\pm$ 0.19)** | **78.13 ($\pm$ 0.23)** | **95.92 ($\pm$ 0.15)** | **78.97 ($\pm$ 0.18)** |

robustness of EE-CBM in scenarios with fewer concepts, as it effectively mitigates the loss of information caused by the reduced concept set. The energy-based pathway and MMD loss in EE-CBM enable efficient utilization of the available concepts, maintaining significant performance even in constrained settings. These results highlight EE-CBM's ability to address the information bottleneck effectively.

# G  APPENDIX: ADDITIONAL RESULTS ON THE CHEXPERT DATASET

Table 8: AUC-ROC Performance Comparison on the CheXpert Dataset. Comparison of AUC-ROC scores (mean ± standard deviation) for various methods, demonstrating the performance of EE-CBM compared to other baseline models.

| Methods | AUC-ROC |
|---|---|
| Bool-CBM | 76.09 ($\pm$1.04) |
| Fuzzy-CBM | 74.14 ($\pm$1.14) |
| CEM | 76.68 ($\pm$0.70) |
| Prob-CBM | 70.45 ($\pm$1.27) |
| ECBM | 78.32 ($\pm$0.93) |
| Coop-CBM | 61.82 ($\pm$0.60) |
| Ours (EE-CBM) | **78.74 ($\pm$ 0.82)** |

To further substantiate our claims, we conducted additional experiments on the CheXpert dataset, evaluating model performance using the AUC-ROC metric. As shown in Table 8, our proposed model (EE-CBM) achieves the highest AUC-ROC score (78.74) with a competitive uncertainty measure ($\pm$ 0.82), outperforming other baseline methods, including ECBM (78.32 ± 0.93) and CEM (76.68 ± 0.70).

These results highlight the robustness of EE-CBM in capturing meaningful concept representations and improving task performance while maintaining reliable uncertainty quantification. This additional evidence further supports the effectiveness of the concept probability branch in addressing the challenges of medical datasets like CheXpert. We include this analysis to provide a more comprehensive evaluation of our model's capabilities.

# H  APPENDIX: EXPLANATION OF EE-CBM'S ROBUSTNESS TO BACKGROUND SHIFTS

To provide further clarity on EE-CBM's improved generalization to background shifts, we elaborate on the mechanisms that enable this robustness. EE-CBM's ability to focus on concept-specific features rather than spurious correlations with background elements is a key factor in its performance. This is achieved through two critical components:

**Energy-Based Pathway:** The energy-based pathway estimates concept probabilities by capturing the intrinsic properties of target concepts, effectively reducing reliance on background information. This probabilistic approach ensures that the model focuses on meaningful features related to the task.

**MMD Loss:** The Maximum Mean Discrepancy (MMD) loss enforces structured separation in the concept space. It clusters similar concepts together while pushing distinct concepts apart, creating a latent space organization that maintains focus on primary object features, even under varying background conditions.

These components work together to enhance EE-CBM's generalization ability, as evidenced by the experimental results discussed in Section 4.1. This explanation provides a theoretical basis for the robustness of EE-CBM to background variations.

## I   APPENDIX: DETAILED EXPLANATION OF INTERVENTION EXPERIMENTS

Intervention experiments in concept bottleneck models involve modifying the model's predicted concept values to study their impact on both the final predictions and the individual concepts themselves. This approach allows us to evaluate how effectively the model handles corrections to its concept predictions, which can be especially relevant in applications requiring high interpretability, such as medical diagnostics or environmental monitoring.

In these experiments, interventions are applied during the test phase, where specific predicted concepts are replaced with their corresponding ground-truth values. This process simulates a scenario in which users or domain experts identify and correct potentially inaccurate concept predictions. By introducing these modifications, we assess how adjustments to one or more concept values influence both downstream classification accuracy and related concepts.

By focusing on test-time interventions, these experiments demonstrate the model's robustness and responsiveness to corrections. This highlights the practical utility of concept-based interpretability in refining predictions without altering the training process, emphasizing the value of this approach in real-world applications.

In the experiments, random concepts were selected with probabilities ranging between 0.0 and 1.0, and their values were replaced with the corresponding ground-truth values. This methodology follows the same approach as used in CEM (Espinosa Zarlenga et al., 2022) and ECBM (Xu et al., 2024), ensuring consistency across the compared methods.

