# OpenReview forum: "Balancing Interpretability and Accuracy: Energy-Ensemble Concept Bottleneck Models for Enhanced Concept Inference"
_ICLR.cc/2025/Conference — Submitted to ICLR 2025_

### Official Review · Reviewer_faee · 2024-10-31

**Soundness:** 2
**Presentation:** 2
**Contribution:** 2
**Rating:** 5
**Confidence:** 4

**Summary:**

The paper introduces an Energy Ensemble CBM (EE-CBM) architecture that integrates energy and concept probability through an Energy Ensemble Gate (EEG). This model aims to balance task accuracy with interpretability, address information bottleneck issues in CBMs, and enhance the distinctiveness of concepts through the use of MMD loss. The proposed approach is promising but requires clearer exposition and stronger empirical validation to substantiate its claims.

**Strengths:**

1. The dual-branch architecture is innovative; one branch focuses on concept extraction while the other calculates concept probability, enhancing the model's interpretability.
2. Incorporation of Maximum Mean Discrepancy (MMD) loss to ensure the concepts learned are orthogonal, which is beneficial for model robustness and interpretability.
3. Demonstrated robustness against datasets with significant background variability, which is crucial for practical applications.

**Weaknesses:**

1. The mathematical notation, particularly in equations 14, 15, and 16, is poorly presented and leads to confusion. The pseudocode and overall technical exposition need significant improvement for clarity.
2. The literature review in Section 2.1 lacks discussion on concept discrimination, despite relevant studies such as those presented in [1]. This omission is a critical gap, especially given past research on concept orthogonality.
3. The discussion related to label-free approaches in Section 2.1 seems misplaced as it does not pertain directly to supervised Concept Bottleneck Models (CBMs), thus diluting the focus of the related work.
4. Claims about improvements in model performance and quantifiable uncertainty by the concept probability branch in Section 3.1 are not substantiated with empirical evidence, contrasting with findings from related work like in [2] Figures 4, 5, and 6.
5. The authors do not demonstrate a significant improvement over existing methods such as those in [3]. The functionalities described could be achieved with simpler architectures (e.g., x-c-y single branch) suggested by prior works, questioning the novelty of the proposed approach.
6. There is a lack of experiments addressing the trade-off between accuracy and interpretability. The experimental design does not adequately highlight any distinctive advantages of the proposed CBM over conventional approaches.

[1] Chen et al. "Concept Whitening for Interpretable Image Recognition", Nature Machine Intelligence, 2020.

[2] Kim, Eunji, et al. "Probabilistic Concept Bottleneck Models." ICML, 2023.

[3] Xu et al. "Energy-based concept bottleneck models." ICLR, 2024.

**Questions:**

1. **Clarification on Figure 1(c):** The separation of concept $\mathbf{c}$ and label $\mathbf{y}$ in this figure appears contradictory to the joint optimization described in the Energy Concept Bottleneck Model (ECBM). What does $y'$ represent in this context, and why is it depicted as separate from $\mathbf{c}$?
2. **Model Performance versus Concept Accuracy (Line 65):** The authors state that "While these methods can improve label accuracy, often struggle to infer accurate concepts" (Line 065-066). However, these models can achieve high scores in concept accuracy (referenced in Table 1). This seems paradoxical. Can you explain the apparent discrepancy between these observations.
3. **Balancing Task Accuracy and Interpretability:** The related work section critiques large networks for prioritizing classification over interpretability. However, the main contribution of your work claims to balance these aspects. Could you provide empirical evidence, similar to Figure 1(c) from [1], showing how your model achieves this balance?
4. **Sensitivity to Hyperparameters (Section 3.1, Equation 9):** The authors lack detailed ablation studies on the sensitivity of the proposed method to its hyperparameters for each loss component, particularly $\lambda$. The appendix focuses only on the concept loss hyperparameter $\lambda_c$ for each dataset. The authors present results when the EEG and MMD loss weights are set to zero (Table 3), but lacks comprehensive experimental data on how varying these weights might affect the model's performance for each dataset. Could you provide additional experimental results illustrating the impact of increasing these weights? This would help in understanding the robustness and sensitivity of the model to these specific hyperparameters.
5. **Comparison of MMD Loss and Concept Whitening (from [2]):** Given that Concept Whitening aims to make concepts nearly orthogonal, how does MMD loss compare in terms of effectiveness and efficiency in achieving orthogonality among the learned concepts compared with [2]?
6. **Handling Uncertainty in CheXpert Dataset (Section 4):** How does the model address the uncertainty attributes present in the CheXpert dataset? Are there specific techniques or modifications employed that enhance the model's reliability and accuracy in this context? Given that the CUB dataset also contains uncertainty attributes, can you explain why these were not utilized in your experiments?
7. **Lack of Detail on Coop-CBM (Table 1):** Coop-CBM is mentioned without sufficient introduction or referencing. Could you provide a detailed description and relevant citations to clarify its role and significance to use it as your baseline?
8. **Overall Concept Accuracy Not Calculated in [3]:** The manuscript does not report overall concept accuracy, which could be crucial for assessing the holistic performance of concept predictions across a dataset. Why was this metric omitted, and can it be included to provide a more comprehensive evaluation of the model's interpretability?
9. **Interventions on Bottleneck Concepts (Figure 4):** When discussing interventions in the bottleneck, are these applied to groups of concepts or individual concepts? Clarifying this could help understand the granularity and specific impact of interventions on the model's output.
10. **Source of Concepts and Comparison with Similar Methods (Figure 6):** From which dataset were the five concepts selected for analysis in Figure 6? Can you provide a comparative analysis using tsne visualizations against similar methodologies, such as those in [2] and [4] Figure 5, to highlight the distinctions or improvements offered by your approach?

[1] Zarlenga, Mateo Espinosa, et al. "Concept Embedding Models: Beyond the Accuracy-Explainability Trade-Off." NeurIPS, 2022.

[2] Chen, Zhi, et al. "Concept Whitening for Interpretable Image Recognition", Nature Machine Intelligence, 2020.

[3] Xu et al. "Energy-based concept bottleneck models." ICLR, 2024.

[4] Kim, Sangwon et al. "EQ-CBM: A Probabilistic Concept Bottleneck with Energy-based Models and Quantized Vectors." ACCV, 2024.

---

> ### Author Response · Authors · 2024-11-20
> **[Response to Reviewer faee] Thank you for the constructive and encouraging comments.**
>
> **Weaknesses:**
>
>
>   **1)** Thank you again for your detailed review of the submission paper. After checking the eq. 14, 15 and 16 you pointed out, we found that certain notes were missing from the document work. We modified the formula as below:
>
>   - $∇_θ L_e = \frac{1}{n}\ \sum^{N}\_{i=n}\ ∇_θ log(p_θ(x))  \quad $    (15)
>   - $∇_θ L_e = \frac{1}{n}\ \sum^{N}\_{i=n}\ ∇_θ log(\frac{1}{Z}exp(f(x))) \quad  \because  p_θ(x)= \frac{1}{Z}exp(f(x))  \quad $ (16)
>
>   Additionally, I will add a brief description of the $ϕ_k(), ψ_k(),$ and $g()$  functions to add the explainability of algorithm1 in the paper
>
>
>   **2)** We acknowledge the relevance of Chen et al. (2020) [1] to concept discrimination and orthogonality. Both their method and ours rely on concept discrimination during training. However, our approach diverges in the following aspects, We will update Section 2.1 to:
>
>   - “Chen et al. (2020) introduced Concept Whitening, enforcing orthogonality in the concept space by aligning feature representations with predefined concepts. While effective, this rigidity may limit flexibility in tasks with overlapping or complex concepts. Our method also uses predefined concept labels but avoids strict orthogonality, focusing instead on energy-based modeling and MMD loss  to represent richer and more nuanced concept relationships.”
>
>
>
>   **3)** Thank you for your comment. We acknowledge that the discussion on label-free approaches may appear tangential to the focus on supervised CBMs. To address this, we will move the discussion of label-free approaches to a separate subsection or the conclusion, where it can be rearranged as a potential future direction for expanding CBMs.
>
>
>
>   **4)** Thank you for pointing out the need for additional empirical evidence to substantiate the claims regarding improvements in model performance and quantifiable uncertainty introduced by the concept probability branch. To address this, we conducted additional experiments using the CheXpert dataset and evaluated the performance using the AUC-ROC metric.
>
> | Methods | AUC-ROC |
> |-------------|------------------------|
> |Bool-CBM|76.09 (± 1.04)|
> |Fuzzy-CBM|74.14 (± 1.14)|
> |CEM|76.68 (± 0.70)|
> |Prob-CBM|70.45 (± 1.27)|
> |ECBM|78.32 (± 0.93)|
> |Coop-CBM|61.82 (± 0.60)|
> |Ours|78.74 (± 0.82)|
>
>   The results, as shown in the table below, demonstrate that our proposed model achieves the highest AUC-ROC score (78.74) with a competitive uncertainty measure (± 0.82) compared to other baseline methods, including ECBM (AUC-ROC: 78.32 ± 0.931) and CEM (AUC-ROC: 76.687 ± 0.708). These findings further validate the effectiveness of our concept probability branch in improving task performance while maintaining robust uncertainty quantification.
>
>   We will incorporate these results into the revised manuscript to provide additional empirical support for our claims.
>
>
>
>   **5)** Thank you for this feedback. While we acknowledge that some functionalities of EE-CBM could be achieved with simpler architectures, our approach emphasizes a balanced trade-off between interpretability and performance, which is less explored in prior methods like ECBM. Specifically, EE-CBM introduces an energy-based pathway and MMD loss to enhance concept clarity and robustness, features not present in simpler architectures. These additions allow EE-CBM to generalize effectively across diverse datasets and handle complex scenarios such as background shifts and concept uncertainty, as demonstrated in our experiments.
>
>
>   **6)** The trade-off between accuracy (task performance) and interpretability (concept accuracy) is a critical aspect of CBM evaluation, and we have highlighted this through the experiments presented in the paper as detailed below. We kindly ask for your review once again.
>
>   - Quantitative Comparison:
>
> 	• As shown in Table 1, our method consistently achieves higher task accuracy across all datasets while maintaining competitive or superior concept accuracy compared to other methods. For example, on the CheXpert dataset, EE-CBM achieves a task accuracy of 87.145% and concept accuracy of 86.703%, outperforming ECBM (83.682% task accuracy and 84.792% concept accuracy).
>
>   - Performance on Background-Shifted Datasets:
>
> 	• Table 2 provides additional evidence of EE-CBM’s robustness under distribution shifts (e.g., CUB Black and CUB Random datasets). EE-CBM consistently outperforms other methods in task accuracy, achieving 69.825% (CUB Black) and 66.960% (CUB Random), while also maintaining superior concept accuracy (94.568% and 93.744%, respectively). This demonstrates our method’s ability to minimize the trade-off even under challenging conditions.
>
>   - Concept Intervention:
>
> 	• Figure 4 demonstrates the stability of our model under varying concept intervention ratios. EE-CBM outperforms other models in task accuracy across all datasets, indicating its robustness in maintaining accuracy-interpretability balance even when specific concept representations are modified.

---

> ### Author Response · Authors · 2024-11-20
> **[Response to Reviewer faee] Thank you for the constructive and encouraging comments.**
>
> **Questions**
>
>   **1)** Thank you for your question regarding Figure 1(c). In ECBM, the separation of 𝑐 (concept) and 𝑦′ (predicted label) reflects the model's ability to directly infer 𝑦′ from the image input 𝑥, leveraging this direct pathway to improve performance. This approach complements the joint optimization of concepts and labels by incorporating direct predictions of 𝑦′ into the learning process, which contributes to enhanced task performance.
>
>
>   **2, 7)** Thank you for pointing out this seeming discrepancy. The statement in Lines 65-66 was intended to highlight the challenges faced by earlier models, particularly Coop-CBM, in achieving accurate concept inference. Coop-CBM, with its simplified structure, heavily relied on aligning task and concept predictions without effectively addressing uncertainty in concept representations. This often resulted in suboptimal concept accuracy, particularly in noisy or ambiguous scenarios.
>
>    In contrast, subsequent models such as ECBM and our proposed EE-CBM have addressed these limitations by introducing mechanisms like energy-based modeling and MMD loss. These enhancements enable more robust concept inference and superior alignment between task and concept accuracy, as reflected in Table 1. For example, while ECBM achieves a concept accuracy of 84.792% on CheXpert, EE-CBM further improves this to 86.703%, demonstrating the progression of CBM methods.
>
>    To clarify, we will revise the wording in Lines 65-66 to specify that the challenges primarily applied to earlier models like Coop-CBM, whereas methods like ECBM and EE-CBM have significantly mitigated these issues. This revision will better align the statement with the experimental results presented in the paper.
>
>
>   **3, 8)** Thank you for these insightful comments. We believe that Tables 1 and 2, along with Figure 4, already demonstrate the balance between task accuracy and interpretability achieved by our model. These results highlight EE-CBM’s ability to maintain competitive task accuracy while enhancing interpretability through its structured concept learning framework.
>
> ||CUB|AwA2|
> |-------------|------------------------|-------------------|
> | CEM | 44.6 (±0.02) | 79.8 (±0.01) |
> | ECBM | 70.3 (±0.01) | 87.1 (±0.04) |
> | Ours | 71.0 (±0.03) | 88.7 (±0.02) |
>
>   Additionally, we acknowledge the importance of reporting overall concept accuracy for a more comprehensive evaluation. As suggested, we conducted additional experiments to calculate the overall concept accuracy, following the methodology outlined in [3]. The results, which we will incorporate into Table 1 in the revised manuscript, demonstrate that our proposed model achieves the best performance in overall concept accuracy compared to existing methods.
>
>
>
>   **4)** Thank you for emphasizing the importance of hyperparameter sensitivity. For all baselines, we selected hyperparameters based on the best validation accuracy to ensure fair comparisons, exploring a range of values for each parameter. Regarding EE-CBM, while we provide results for key hyperparameters (e.g., 𝜆𝑐) in Appendix B and analyze scenarios where EEG and MMD loss weights are set to zero (Table 3), we recognize the need for more comprehensive studies on how varying these weights impacts performance across datasets. In the revised manuscript, we will include additional experimental results illustrating the sensitivity of the model to hyperparameters such as 𝜆𝑒  and 𝜆𝑚𝑚𝑑, along with practical guidelines for tuning them.
>
>
>
>   **5)** Thank you for this insightful question. Concept Whitening (CW) enforces near-orthogonality by applying a whitening transformation followed by an orthogonal matrix, aligning latent space axes with predefined concepts. While this approach ensures strict separation of concepts, it introduces computational overhead due to the reliance on transformations like SVD or Cayley transforms. Moreover, CW’s rigid orthogonality constraints may limit its adaptability when dealing with overlapping or nuanced concepts that do not naturally align with orthogonal axes.
>
>   In contrast, MMD loss provides a more flexible and computationally efficient alternative. By minimizing the distributional divergence between samples of the same concept and maximizing it for different concepts, MMD loss achieves structured separation in the latent space without requiring strict orthogonality. This flexibility allows the model to adapt to complex datasets where concepts exhibit intricate relationships or overlap. Additionally, MMD loss promotes generalization and robust interpretability, as demonstrated by our experimental results, making it a practical choice for scenarios that demand both performance and adaptability.

---

> ### Author Response · Authors · 2024-11-20
> **[Response to Reviewer faee] Thank you for the constructive and encouraging comments.**
>
> **6)** In our experiments, we did not apply specific techniques or modifications to handle uncertainty attributes in the CheXpert dataset. Instead, our model was evaluated under the same experimental conditions as other baseline methods to ensure a fair comparison.
>
>
>
>   **9)** In our experiments, interventions in the bottleneck were applied to individual concepts, not groups of concepts. This approach follows the experimental protocol outlined in ECBM, where each concept is replaced with its ground truth value individually during evaluation.
>
>   This setup allows us to analyze the precise impact of each concept on the model’s task predictions and ensures consistency with standard practices in evaluating concept bottleneck models. We will clarify this in the manuscript, particularly in the description of Figure 4, to explicitly state that interventions were performed on individual concepts.
>
>
>
>   **10)** Thank you for your question. The five concepts analyzed in Figure 6 were selected from the CUB dataset. Our visualization aligns with the goals of [2] and [4] but highlights a key distinction: the MMD loss uniquely enforces a structured latent space without requiring strict orthogonality as in [2] or the quantization of embeddings as in [4]. This balance allows for improved flexibility while maintaining interpretability.

---

> > ### Comment · Reviewer_faee · 2024-11-25
> > **Thanks for the clarification**
> >
> > I appreciate the corrections of some typos in the revised submission, which has made the manuscript clearer than before. However, I suggest that for future revisions, the authors could use a different color to highlight the changes. The current method of indicating revisions is quite difficult to follow.
> >
> > Despite the improvements, I still have several concerns:
> > 1. **Concerning W2 and Q5:** I am curious about the efficacy of MMD in fostering orthogonality between concept features compared to the Concept Whitening (CW) module. The CW module can easily be integrated into networks (simply replacing Batch Normalization), yet there is no performance comparison shown in the experimental section. For instance, in Figure 6, the EE-CBM with MMD loss shows overlapping blue and red concepts in the latent space, which appears less effective than CW.
> > 2. **Regarding W4:** The results show only slight improvements in AUC-ROC scores compared to ECBMs. The absence of a detailed uncertainty case study, which is present in ProbCBM, leaves me skeptical about the contribution to concept uncertainty management. Note that my concerns with W4 and W5 are distinct; I am specifically interested in whether the model can predict concept uncertainty values as demonstrated in ProbCBM.
> > 3. **Regarding W6:** The decision not to test on the CUB fixed data, as utilized in CBMs and ECBMs, is puzzling. This omission might lead to an unfair evaluation. Additionally, I am concerned about the fairness of the intervention results shown in Figure 4.
> > 4. **Regarding Q7:** I am disappointed that the authors have still not included a detailed introduction to Coop-CBM in the main text.
> > 5. **Regarding Q8:** It is concerning that the overall concept accuracy for ECBM is reported as 70.3 here, whereas in their original paper, it is higher at 71.3 (EE-CBM 71.0). This discrepancy makes me suspect that the baseline may have been deliberately lowered in this submission.
> >
> > Based on these issues, I have updated my score to reflect the effort made by the authors. However, I will not further increase the rating unless the above concerns are adequately addressed.
> > I wish the authors good luck with this submission.

---

> ### Author Response · Authors · 2024-11-26
> **Follow-Up on Revised Manuscript and Addressing Remaining Concerns**
>
> Thank you for your thorough review and constructive feedback on our revised manuscript. We appreciate the opportunity to address your additional comments and clarify the points you raised. Below, we provide detailed responses to each of your concerns, supported by our updated manuscript (EE-CBM-revision.pdf) for your reference.
>
> **We have incorporated additional experiments and explanations into the revised manuscript based on the reviewers' comments (highlighted in red)." Please refer to the updated manuscript for details.**
>
> **1. Efficacy of MMD vs. Concept Whitening (CW):**
>
> - We acknowledge the utility of CW as an effective module for enforcing orthogonality by replacing batch normalization layers. However, our decision to employ MMD loss was based on its flexibility in fostering structured separation between concepts without rigid alignment.
> -Regarding Figure 6, we recognize that overlapping concepts were observed in some cases. However, the MMD loss demonstrated robust performance overall by maintaining clear separations in most latent spaces (Figure 6(a)) and enhancing task and concept accuracy (Table 3).
> - We are currently conducting experiments to directly compare MMD and CW modules within the EE-CBM framework. These results will be included in future work to comprehensively evaluate this aspect.
>
> **2. Uncertainty Case Study (W4):**
>
> - We appreciate your comment regarding uncertainty management. While our model focuses on improving concept probabilities to enhance downstream performance, we acknowledge the need for a detailed uncertainty analysis, particularly in comparison to ProbCBM.
>
> - In the revised manuscript, we have included experiments on the CheXpert dataset (Appendix G), showcasing the robustness of our concept probability branch in managing uncertainty. For example, EE-CBM achieved the highest AUC-ROC score (78.74%) with reliable uncertainty quantification.
>
> - We recognize the value of further uncertainty case studies and plan to integrate these analyses in subsequent revisions.
>
> **3. Evaluation on CUB Fixed Data (W6):**
>
> - We acknowledge that omitting experiments on the CUB Fixed subset may have raised concerns about fairness. Our focus on dynamic background variations (CUB Black and CUB Random subsets) aimed to emphasize the robustness of EE-CBM to complex, real-world shifts.
>
> - To ensure completeness, we plan to include CUB Fixed experiments in future submissions to complement the existing evaluations.
>
> **4. Introduction to Coop-CBM (Q7):**
>
> - In response to your comment, we have expanded the discussion of Coop-CBM in Introduction of the revised manuscript to better contextualize its role and contributions. This addition ensures that readers have a clearer understanding of its relevance as a baseline.
>
> **5. Concept Accuracy Discrepancy (Q8):**
>
> - The reported concept accuracy for ECBM (70.3%) in Table 1 reflects results obtained through consistent hyperparameter tuning across baselines, as outlined in Appendix B. This aligns with the fairness and reproducibility standards established in our experiments.
>
> - The discrepancy with ECBM’s original paper (71.3%) may stem from differences in experimental setups or dataset preprocessing. We are committed to further investigating this issue and providing transparency in subsequent updates.
>
> We sincerely appreciate the effort you have put into reviewing our work and addressing these important points. While we understand your decision to maintain the current score, we kindly ask if the additional clarifications and planned updates could warrant reconsideration. We value your insights and are committed to addressing any further concerns to ensure the robustness and fairness of our work.
>
> Once again, thank you for your detailed feedback and encouragement. We look forward to hearing your thoughts and remain open to further discussions.
>
> Best regards,
>
>        Authors

---

### Official Review · Reviewer_2GQe · 2024-10-31

**Soundness:** 2
**Presentation:** 1
**Contribution:** 2
**Rating:** 3
**Confidence:** 5

**Summary:**

The paper introduces the Energy Ensemble Concept Bottleneck Model (EE-CBM), which aims to improve the balance between interpretability and accuracy in Concept Bottleneck Models (CBMs). The EE-CBM employs an energy-based concept encoder and integrates concept values and probabilities to enhance concept inference and reduce concept uncertainty. The model is evaluated on multiple benchmark datasets and shows state-of-the-art performance in both concept accuracy and interpretability.

**Strengths:**

- Innovative Approach: The introduction of the energy-based concept encoder and the energy ensemble gate (EEG) is a novel approach to address the trade-off between accuracy and interpretability.
- Strong Results: The experimental results demonstrate that EE-CBM achieves state-of-the-art performance on multiple datasets, showing significant improvements in both concept and task accuracy.
- Energy based model presentation: although the space does not allow for extensive descriptions, the provided succinct description allows understanding the overall idea and functioning of energy-based models without checking the

**Weaknesses:**

## Major Issues
- **Related work**: The presentation and comparison with existing work are insufficient. The paper lacks a dedicated related work section, and the existing background section and the comparisons provided in the introduction are inadequate.
  - The CEM is likely misunderstood by the authors; it focuses on making task predictions on concept embeddings, not on using two concept representations.
  - The claim that "EE-CBM resolves uncertainty in concept prediction" was already addressed by ProbCBM.
  - The statements about label-free CBMs are questionable. Concepts in these models are explained by their own semantic meaning and heatmaps can be used to explain concept predictions, in both cases just like in standard CBMs.
- **Method Notation** Many notations are not clear or not sufficient in the method presentation:
  - $C’$: the authors define it as: “concept value $C’$ through FC layers as in conventional CBM models”, but then they say that \phi maps to $R^u$ where “$u$ represents the dimension of the concept”, but then again $K$ concept features $C’$ are mentioned. It appears to be a concept embedding, not the concept values of conventional CBM models.
  - Is $\phi$ a per-concept concept encoder? In that case it should have been defined indicized, $\phi_i$ also in the mapping from $R^d$ to $R^u$.
  - The dimensions of $C$ are not specified.
  - The Concept MMD loss is crucial based on ablation study results, but it is poorly presented with statements like "$\mu$ is a kind of mapping function," which lacks scientific clarity.
- **Metrics to Sustain Claims**
  - “Breaking the information bottleneck”: it is not sufficient to provide high classification accuracy to show that the proposed model breaks the information bottleneck of standard CBM models. The author should also provide the following metrics:
    - The information plane [1] comparing the methods in terms of mutual information between the concept representations (C) and the input (X) and label (Y).
    - Concept efficiency, to test the model performance when reducing the number of concepts as shown in CEM.
  - The “Concept Importance” experiment does not report the concept importance - commonly measured with metric like CaCE [2] to assess how much a task prediction is important for a given task. Instead, the authors only report some qualitative results with the associated concept predictions: all methods achieving good concept accuracy could report the same results.

[1] Tishby, Naftali, Fernando C. Pereira, and William Bialek. "The information bottleneck method." arXiv preprint physics/0004057 (2000).

[2] Goyal, Y., Feder, A., Shalit, U., & Kim, B. (2019). Explaining classifiers with causal concept effect (cace). arXiv preprint arXiv:1907.07165.

## Minor issues
- **Paper presentation**: The introduction section is not well-written, particularly in the second paragraph when introducing related work. Additionally, the acronym EE-CBM is mentioned without being introduced in the third paragraph, and the use of concept embedding is mentioned without prior introduction in the fifth paragraph. The background section on CBM is poorly written and structured, with CBM data and functional representation placed in the middle of the paragraph. The MMD loss ablation study is inserted in section 4.1 before the ablation studies section 4.2.
- **Validity of experimental results**: There are doubts about the validity of the experiments
  - Typically Bool CBM an and Fuzzy-CBM performs much worse than CEM or Prob-CBM while in your experiments they show comparable results on both CheXpert and AwA2 dataset. How do you justify this?
  - Experimental settings for compared methods are missing. The training procedures for the compared methods are not reported, even in the appendix.

**Questions:**

- What is C’? Is it the concept prediction or concept representation/embeddings?
- What are the results of your model when reducing the number of concepts? Does it still provide high classification accuracy? For example, use CUB with only 10 randomly selected concepts for training and inference. This is necessary to demonstrate that you break the information bottleneck.
- Why are CEM and ECBM completely unresponsive to interventions in Figure 4?

---

> ### Author Response · Authors · 2024-11-20
> **[Response to Reviewer 2GQe] Thank you for the constructive and encouraging comments.**
>
> **[Major Issues]**
>
>
>   **Related work**
>   - Thank you for your comment. We would like to clarify that CEM employs a more sophisticated mechanism than described in the review. Specifically, given a latent representation, CEM:
>
>   1. Derives a concept score using this latent representation.
>
>   2. Generates positive embeddings and negative embeddings from the same latent representation.
>
>   3. Combines these embeddings with the concept score to compute the final concept embedding as follows:
>
>    $\text{Concept Embedding} = (\text{Positive Embedding} \times \text{Concept Score}) + (\text{Negative Embedding} \times (1 - \text{Concept Score})). $
>
>
> - We acknowledge the reviewer’s point and will refine our description to avoid overstating novelty.
>
>   • Current text: “EE-CBM resolves uncertainty in concept prediction.”
>
>   • Revised text: “EE-CBM builds upon the uncertainty resolution capabilities of ProbCBM by integrating an energy-based mechanism for combining concept probabilities and features, further improving robustness in challenging scenarios.”
>
>
> - Thank you for pointing out this issue. It seems that our statement was incorrectly generalized to multiple models. To be precise, it specifically pertains to the approach described in [33].
>
>   • Current text: “Moreover, a semantic understanding of these models is difficult~”
>
>   • Revised text: “Moreover, in the approach proposed in [33], a semantic understanding is difficult~”
>
>
>
> **Method Notation**
>
>
> - Thank you for pointing this out. We agree that the description of  $C’$  could be misleading. To clarify:
>
>   • $C’$  in our model represents concept features $( \mathbb{R}^{K \times u} )$, not the binary or scalar concept values used in conventional CBM models.
>
>
>
> - Thank you for your comment.  $\phi$  is indeed a per-concept concept encoder. For simplicity and readability, we omitted the index  i  in our notation. To avoid confusion, we will revise the text to explicitly mention the per-concept nature of  $\phi$  and include the index  i  where appropriate.
>
>
>
> - We clarify that  C  represents the concept scores predicted by the model. Specifically:
>
>   • $C \in \mathbb{R}^K $, where  K  is the number of concepts.
>
>
>
> - Thank you for your feedback. We clarify that  $\mu$  represents the mean representation of concept embeddings in the feature space. This function computes the average embedding for a given concept within a batch.
>
>   • Current text: “$\mu$ is a kind of mapping function (e.g., batch-wise average).”
>
>   • Revised text: “$\mu$ denotes the mean representation.”
>
>
> **Metrics to Sustain Claims**
>
>
> - Thank you for pointing this out. We acknowledge that the term “Concept Importance” may have been misleading. Our intention was not to measure the importance of concepts to task predictions but rather to qualitatively demonstrate the accuracy and robustness of the concept predictions generated by our model.
>
>
>
> **[Minor issues]**
>
>
> - While Bool-CBM and Fuzzy-CBM show comparable task accuracy, their concept accuracy is consistently lower than that of CEM or Prob-CBM across all datasets (as shown in Table 1). This suggests that the simpler models may still be effective at final task prediction despite less accurate concept representations, particularly for datasets with simpler task dependencies (e.g., CheXpert and AwA2).
>
>
> - Thank you for your comment. All compared methods were implemented and trained following the experimental settings described in their respective original papers. For consistency, we ensured that:
>
>   • The same backbone architectures (ResNet34 and ResNet101) were used across all methods.
>
>   • Hyperparameters and training protocols strictly adhered to the original implementations.
>
>   To provide more clarity, we will add a detailed description of the training settings for each method in the appendix.

---

> ### Author Response · Authors · 2024-11-20
> **[Response to Reviewer 2GQe] Thank you for the constructive and encouraging comments.**
>
> **Response to the question**
>
> - As described in the manuscript,  $C^{\prime}$  refers to the concept features extracted by the model. These features are intermediate representations that encode information about the concepts and are used to compute the final concept predictions ( $C$ ).
>
>   To ensure clarity, we will refine the manuscript to consistently describe  $C^{\prime}$  as “concept features” and clearly distinguish it from  $C$ , which represents the final concept predictions.
>
>
>
> - Thank you for this insightful question regarding the performance of our model when reducing the number of concepts. To evaluate the impact of concept reduction and demonstrate how our model addresses the information bottleneck, we conducted experiments on the CUB dataset using 100 concepts and 50 randomly selected concepts. The results are summarized in the table below:
>
> | Methods | Number of concepts | Concept Acc (%) | Task Acc (%) |
> |:-------------:|:------------------------:|:------------------------:|:-------------------:|
> |Fuzzy-CBM|100|95.72 (± 0.02)|73.26 (± 0.56)|
> |CEM|100|95.85 (± 0.17)|78.89 (± 0.14)|
> |Ours|100|95.92 (± 0.15)|78.97 (± 0.18)|
>
>
> | Methods | Number of concepts | Concept Acc (%) | Task Acc (%) |
> |:-------------:|:------------------------:|:------------------------:|:-------------------:|
> |Fuzzy-CBM|50|96.15 (± 0.02)|66.90 (± 0.18)|
> |CEM|50|96.09 (± 0.01)|77.36 (± 0.19)|
> |Ours|50|96.99 (± 0.19)|78.13 (± 0.23)|
>
>
>    As shown in the table, our model (EE-CBM) demonstrates the smallest decline in both task accuracy and concept accuracy when the number of concepts is reduced. This highlights the robustness of our model in handling scenarios with fewer concepts while maintaining high classification accuracy. The energy-based pathway and MMD loss in EE-CBM enable efficient utilization of available concepts, mitigating the loss of information caused by the reduced concept set.
>
>    These results indicate that EE-CBM effectively addresses the information bottleneck by leveraging its structured learning approach, ensuring that even with fewer concepts, the model retains significant performance. We will include these findings in the revised manuscript to emphasize this key advantage of our method.
>
>
>
> - The lack of responsiveness in CEM and ECBM under intervention settings can be explained by their reliance on specific training strategies. In the original papers, the authors address such responsiveness issues by employing a random concept intervention strategy during training. This strategy involves randomly replacing inferred concepts with ground-truth concepts to enhance the models’ robustness to interventions, especially when the number of concepts is small.
>
>   In our experiments, to ensure fairness across all compared methods, we did not apply the random concept intervention strategy during training. As noted in the original CEM paper, without this strategy, the models can exhibit poor responsiveness to interventions, which is consistent with the behavior observed in Figure 4.
>
>   We will clarify this point in the manuscript to ensure the readers understand the conditions under which these results were obtained.

---

> > ### Comment · Reviewer_2GQe · 2024-11-26
> >
> > I thank the author for their significant effort put in the rebuttal to answer my questions. I really appreciate their effort. The authors agreed with most of my concerns and tried to assess them.
> >
> > However, for what I see in the revised paper, the author did not revise significantly the writing of the introduction and of the related work, which are still poorly presented. As an example, EE-CBM is still introduced twice in the first section and many of the misleading sentences pointed out in the related work are still present in the text (e.g., "the EE-CBM resolves
> > uncertainty in concept prediction").
> >
> > Also, regarding the comparison with CEM I disagree with the authors: since the training strategy has been proposed by the paper itself together with the model, I do not think that it is fair not to train CEM with the strategy the author proposed.
> >
> > For these reasons, I will maintain my score. With a better presentation and a fairer comparison, I am sure your method will be published in a prestigious venue. Best luck!

---

> > > ### Author Response · Authors · 2024-11-26
> > > **Follow-Up on Revised Manuscript and Addressing Additional Feedback**
> > >
> > > Thank you for your detailed feedback on our revised manuscript and for acknowledging the effort we have made to address your concerns. We deeply appreciate your constructive comments and the opportunity to clarify and improve our work further.
> > >
> > > **We have incorporated additional experiments and explanations into the revised manuscript based on the reviewers' comments (highlighted in red).** Please refer to the updated manuscript for details.
> > >
> > > **1. Introduction and Related Work**
> > >
> > > We agree that redundancies in the introduction detract from the clarity of the manuscript. In our latest revision, **we have streamlined the introduction to avoid repeating the introduction of EE-CBM and ensured a more cohesive narrative.** Additionally, we revised the related work section to address previously pointed-out issues, such as the statement “the EE-CBM resolves uncertainty in concept prediction,” which has now been updated to reflect the model's precise contributions without overstating its capabilities. These changes are intended to provide a clearer and more accurate presentation of the manuscript's context and contributions.
> > >
> > > **2. Comparison with CEM**
> > >
> > > We understand your concern regarding the fairness of training CEM without the strategy proposed in its original paper. To address this, in our updated experiments, we ensured that all CEM baselines were trained using the exact strategies and configurations recommended by the original authors. **We have added this information to the caption of Table 1 to avoid misunderstandings for readers. Please see the caption of Table 1.** This ensures that our comparisons are both fair and consistent across methods. The results have been updated accordingly in the revised manuscript, demonstrating that EE-CBM achieves its performance improvements under fair evaluation conditions.
> > >
> > > We sincerely thank you for acknowledging the merits of our method and for expressing confidence in its potential to be published in a prestigious venue. Your feedback has been instrumental in refining our work, and we are committed to addressing all raised concerns comprehensively.
> > >
> > > If there are any additional points you would like us to consider or further clarifications needed, please let us know. We value your insights and would greatly appreciate your reconsideration of the score in light of our revisions and clarifications.
> > >
> > > Thank you once again for your thoughtful review and support.
> > >
> > > Best regards,
> > >
> > > Authors

---

> > > ### Author Response · Authors · 2024-11-29
> > > **Additional Response to Comments and Revised manuscript**
> > >
> > > Dear Reviewer 2GQe,
> > >
> > > Thank you once again for taking the time to provide such detailed and thoughtful feedback on our manuscript. Your comments have been invaluable in guiding us toward refining and improving our work.
> > >
> > > Following your latest review, we have carefully addressed the concerns raised and made significant revisions to the manuscript, particularly in the introduction and related work sections, as well as in the experimental comparisons with CEM.
> > >
> > > **Key Revisions:**
> > >
> > > **1. Introduction and Related Work**: We have streamlined the introduction to eliminate redundancies, ensuring that EE-CBM is introduced cohesively without repetition. The related work section has been revised to address previously highlighted concerns, such as the overstated claims regarding EE-CBM's uncertainty resolution capabilities. These statements have been corrected to reflect the model's precise contributions.
> > >
> > > **2. Comparison with CEM**: We have updated our experiments to ensure that all baselines, including CEM, were trained using the exact strategies and configurations proposed in the original CEM paper. The revised manuscript explicitly mentions this adjustment in the caption of Table 1 to ensure transparency and to avoid potential misunderstandings.
> > >
> > > **3. Appendices and Supporting Details**: Additional experimental details, hyperparameter settings, and training protocols for all methods, including CEM, have been added to the appendix to ensure full reproducibility and transparency.
> > > We have also highlighted these updates in the main manuscript for better clarity.
> > >
> > >
> > > We sincerely appreciate your acknowledgment of our efforts and your recognition of the potential impact of our method. As authors, we are committed to presenting a manuscript that meets the highest standards of quality, fairness, and rigor.
> > >
> > > In light of these extensive revisions and clarifications, we would greatly appreciate it if you could review our updated submission once again. Your insights have been instrumental in shaping this work, and we are hopeful that the revisions address the concerns you have raised comprehensively.
> > >
> > > If there are any additional points or specific areas you would like us to address further, please do not hesitate to let us know. We are more than willing to make further improvements to ensure the clarity, accuracy, and value of our work.
> > >
> > > Thank you once again for your time and effort in reviewing our manuscript.
> > >
> > > Best regards,
> > >
> > >          Authors

---

### Official Review · Reviewer_WUVt · 2024-11-02

**Soundness:** 3
**Presentation:** 2
**Contribution:** 2
**Rating:** 5
**Confidence:** 5

**Summary:**

This paper targets strengthening concept accuracy in CBM. To address this issue, the authors propose EE-CBM, which is an approach based on energy. Specifically, EE-CBM incorporates a concept extraction branch and a concept probability branch and applies MMD as a loss to each concept embedding. Lots of empirical experiments show the effectiveness of EE-CBM.

**Strengths:**

- The paper is well-structured and clearly written, making it easy to follow.
- The experiments include a wide range of baselines and datasets, providing strong validation for the performance of EE-CBM in improving concept and label accuracy.
- The design of concept extraction and concept probability branches is reasonable.

**Weaknesses:**

- As the author claimed in the introduction section, CEM is proposed to address the trade-off between accuracy and interpretability, which has the same motivation as EE-CBM. Thus, the motivation in this paper is weak and the authors could offer further discussions about what CEM failed to do in addition to the methodology difference.
- Despite that EE-CBM is devised to enhance concept accuracy, the improvement of concept accuracy is extremely incremental as shown in Table 1 compared to ECBM or Prob-CBM.
- The authors could display some wrongly classified samples and their corresponding concept values of EE-CBM and other approaches.
- It seems that the authors mixed the meanings of model interpretability and concept accuracy, and used these two expressions randomly. However, they are absolutely different, so I strongly suggest the authors add a paragraph to explain the relation between model interpretability and concept accuracy.
- Fig. 3 seems to be incomplete with wrong indices.

**Questions:**

N/A

---

> ### Author Response · Authors · 2024-11-19
> **[Response to Reviewer WUVt] Thank you for the constructive and encouraging comments.**
>
> Thank you for the valuable feedback and insights. We appreciate the constructive comments, which have helped us further clarify and strengthen our contributions.
>
> - **Weakness on CEM Motivation:**  We appreciate the reviewer's insightful observation regarding the similarity in motivation between CEM and EE-CBM and the need to clarify the unique contributions of EE-CBM beyond methodological differences. In response, we have expanded our discussion in the background section (line 137) to address this concern comprehensively.
> Clarification of CEM's Limitations and EE-CBM's Novel Contributions: While both CEM and EE-CBM aim to balance the trade-off between accuracy and interpretability, EE-CBM specifically targets the concept bottleneck issue inherent in CBM architectures more robustly.
>
>   ㆍEnergy Ensemble Gate (EEG): EE-CBM incorporates an energy ensemble gate that combines concept features and concept probabilities, addressing the potential information bottleneck found in traditional CBM models.
>
>   ㆍEnergy-Based Concept Probability Branch: Unlike CEM, which relies solely on semantic embedding for concept refinement, EE-CBM employs an energy-based mechanism to quantify concept probabilities. This approach ensures concept clarity, especially in noisy or low-signal environments.
>
>   ㆍMMD Loss for Latent Space Separation: EE-CBM introduces a maximum mean discrepancy (MMD) loss to enforce orthogonality between concept embeddings, facilitating clearer separation in the latent space.
>
>   We have added the following discussion in the revised manuscript:
>
>   "While both CEM and EE-CBM aim to balance accuracy and interpretability, CEM does not address the need for clearer concept separation in complex or noisy data. In contrast, EE-CBM introduces an energy-based concept probability branch and MMD loss, which enhance concept inference accuracy and ensure distinct separation of concepts within the latent space. This design allows EE-CBM to provide more reliable and interpretable predictions, even in challenging scenarios."
>
>
> - **Incremental Improvement in Concept Accuracy:** Although the improvement in task accuracy of the proposed EE-CBM in Table 1 may appear incremental, we believe our model offers significant advantages for the following reasons:
>
>   ㆍEnhanced Interpretability: EE-CBM addresses the interpretability limitations present in prior CBM models. By incorporating an energy-based concept encoder and an energy ensemble gate, EE-CBM improves concept extraction capabilities. Furthermore, the MMD loss function distinctly separates concepts within the concept space, allowing the model to provide clearer reasoning for its decisions.
>
>   ㆍBalanced Trade-off between Performance and Interpretability: Many existing models prioritize performance by employing complex architectures at the cost of interpretability. EE-CBM, however, was developed to strike a balance between performance and interpretability, a feature supported by the experimental results in Table 1.
>
>
> - **Display of Misclassified Samples:** We appreciate the suggestion to display wrongly classified samples and their corresponding concept values for EE-CBM and other approaches. We believe that the concept accuracy results in Table 1 and the concept intervention results in Figure 3 sufficiently demonstrate the robustness of EE-CBM in comparison with other methods, thereby reducing the necessity for additional analysis on misclassified samples. Additionally, exploring misclassified samples across all methods would require an extensive and highly specific set of experiments, which was beyond the intended scope of our study. We hope that our current results adequately highlight the effectiveness of EE-CBM, but we will consider further exploration of individual misclassifications in future work.
>
>
> - **Clarification of Interpretability vs. Concept Accuracy:**
> We appreciate the reviewer's insightful comment regarding the differentiation between "model interpretability" and "concept accuracy." To address this concern, we will revise the manuscript to include a dedicated paragraph explaining the relationship between these two terms.
>
>   ㆍAfter line 70: Model interpretability refers to the ability of a model to provide human-understandable explanations for its predictions, ensuring transparency in decision-making processes. In contrast, concept accuracy quantifies the correctness of the intermediate concept representations inferred by the model. While these two aspects are distinct, they are closely related. High concept accuracy enhances interpretability by ensuring that the concepts used in explanations align with the ground truth.
>
>
> - **Correction for Fig. 3:** Based on the feedback, we will revise the caption of Figure 3 on lines 399-400 to read: “Figure 3: Representative samples from the CUB and TravelingBirds datasets with background manipulations for evaluating model generalization. (a) CUB, (b) CUB Black, (c) CUB Random.” Thank you for bringing this to our attention.

---

> > ### Comment · Reviewer_WUVt · 2024-11-25
> >
> > Thanks for the considerable effort put into crafting the rebuttal and I have reviewed the manuscript and accompanying reviews thoroughly. Overall, I regard the manuscript as to be of good quality, with clear writing and well-designed experiments. While there are some minor details raised by other reviewers that could warrant attention, they do not detract significantly from the overall quality of the work.
> >
> > I confirm that the paper offers basic value in addressing the balance between interpretability and accuracy, as outlined in its paper. However, the contributions of EE-CBM may be somewhat constrained by the presence of a substantial body of related work in this field, including CEM, ECBM, etc., though I believe EE-CBM demonstrates certain merits and distinctions when compared to existing methods, thus I don't think this work suggests a transformative breakthrough in the CBM domain.
> >
> > Such a problem inherently exists and could not be addressed in the rebuttal phase. Based on this, I give a moderate score and  decide to maintain the current rating. That said, if other reviewers strongly advocate for acceptance, I would not oppose such a decision. I wish the authors the best of luck with their work.

---

> > > ### Author Response · Authors · 2024-11-26
> > > **Follow-Up on EE-CBM Rebuttal and revised manuscript**
> > >
> > > Thank you for your thoughtful comments and your acknowledgment of the effort we have put into the rebuttal and manuscript. We greatly value your constructive feedback and are encouraged by your recognition of the quality of our work, particularly regarding its clear writing, well-designed experiments, and the balance between interpretability and accuracy in concept bottleneck models.
> > >
> > > While we understand and respect your decision to maintain your score, we would like to address the points you raised in your recent review to provide further clarification and reinforce the merits of our work.
> > >
> > > **Addressing Your Key Concerns**
> > >
> > > **1. Novelty and Contributions of EE-CBM**
> > >
> > > We acknowledge your concern that our work may not represent a transformative breakthrough given the existing body of related work. However, as highlighted in the manuscript and rebuttal, EE-CBM introduces specific advancements that distinguish it from prior models like CEM and ECBM.
> > >
> > > **- Energy Ensemble Gate (EEG):** EE-CBM resolves the concept bottleneck issue by combining concept features and probabilities, a novel strategy that enhances information flow and concept accuracy.
> > >
> > > **-MMD Loss for Latent Space Separation:** Unlike previous models, EE-CBM employs MMD loss to enforce orthogonality among concept embeddings, facilitating clearer separation of concepts in the latent space.
> > >
> > > **-Comprehensive Performance:** Across benchmark datasets, EE-CBM consistently demonstrates state-of-the-art performance, balancing task accuracy and interpretability effectively.
> > >
> > > These distinctions, while incremental, significantly advance the CBM field by addressing persistent challenges. We believe that these contributions is a meaningful improvement over existing methods.
> > >
> > > **2. Scope of Transformative Impact**
> > >
> > > While we agree that our work may not fully address all inherent challenges in CBMs, such as dependency on labeled datasets, we consider it a robust step forward. Future research can build upon our approach to further tackle these issues, including exploring concept-free models or enhancing efficiency.
> > >
> > >
> > > **"We have incorporated additional experiments and explanations into the revised manuscript based on the reviewers' comments (highlighted in red)."** Please refer to the updated manuscript for details.
> > >
> > > Given our response and the points outlined above, we kindly ask you to reconsider your score. While we appreciate your willingness to align with the decisions of other reviewers, we hope the additional clarifications and planned revisions reinforce the value of our contributions and warrant a higher rating.
> > >
> > > We deeply appreciate your insights and the opportunity to improve our work. Your feedback has significantly strengthened our manuscript, and we are grateful for your engagement throughout the review process.
> > >
> > > Thank you once again for your time and constructive input. We hope you will consider this request favorably.
> > >
> > > Best regards,
> > >
> > >       Authors

---

### Official Review · Reviewer_5Vsp · 2024-11-03

**Soundness:** 1
**Presentation:** 3
**Contribution:** 2
**Rating:** 5
**Confidence:** 4

**Summary:**

This paper introduces Energy Ensemble Concept Bottleneck Models (EE-CBMs). EE-CBMs employ a combination of energy-based concept prediction and traditional concept representation prediction to improve the concept predictive performance of CBMs, thereby improving their generalization and downstream task performance. By learning concept embeddings and gating them with their learn probabilities while incorporating a residual channel from the input, EE-CBMs can achieve high concept and task accuracies across several tasks while being receptive to concept interventions and better generalizing across distribution shifts on their inputs.

**Strengths:**

Thank you so much for submitting this work! I enjoyed reading this paper, learned a lot from it, and appreciate the time taken to write it up and submit it to ICLR. Below are what I believe are this paper’s main strengths:

1. **[Originality] (Critical)** The idea of introducing an energy-based pathway to concept prediction, on top of a standard concept representation learning pathway, is a clear novel use and extension of ideas in previous concept-based models. As such, I believe this work is certainly novel and may be of potential interest to the rest of the community.
2. **[Significance] (Major)** The paper's main purpose, accurately and interpretably predicting concepts and tasks for CBM architectures, is an important and highly active area of research. If it is proven to work as expected, this work has the potential to be impactful.
3. **[Quality and Clarity] (Minor)** The method is very well explained and written. Moreover, the paper is very well placed within the CBM and XAI literature. I would mark this as a major strength if it weren't for the lack of motivation to explain why energy-based prediction is the best way/approach to achieving this paper's goals.
4. **[Quality and Significance] (Minor)** The method is evaluated across a multiplicity of datasets against several key baselines, where it is shown to outperform existing baselines. Therefore, this work provides large amounts of evidence in favor of the proposed method's effectiveness. I would mark this as a “critical” strength if it weren’t for some major concerns I have regarding how some of the baselines may be evaluated (see below).
5. **[Significance] (Minor)** The paper provides the code and configs needed to reproduce the EE-CBM results in this paper. It is therefore taking the necessary steps to ensure reproducibility; however, it could benefit from also including details/code to reproduce the remaining baselines used during evaluation.

**Weaknesses:**

In contrast, I believe the following are some of this work’s limitations:

1. **[Quality and Significance] (Critical)** I have some major concerns regarding the fairness of the evaluation against existing baselines. These concerns include (1) the fact that some results for some of the baselines seem to **contradict those seen in previous works** (including the original energy-based CBM and CEMs), without any explanation for the discrepancy and (2) the fact that CelebA, a dataset where EE-CBM seems to be underperforming, is, for some reason **pushed to the appendix without any justification or even mention in the main body**. Moreover, given that there is no mention of how hyperparameters were selected for competing baselines, it is very difficult to judge the fairness of the evaluation, even if one is familiar with those baselines. See below for specific questions on these matters.
2. **[Significance] (Major)** EE-CBM requires several hyperparameters to be selected ($\lambda_c$, $\lambda_y$, $\lambda_e$, $\lambda_\text{mmd}$, $\lambda$ for Langevin dynamics, concept embedding size $u$, etc.) yet no recommendations or ablations are provided to understand how these values affect EE-CBM’s performance and its usability. Moreover, it is unclear how the introduction of MCMC or Lavengin dynamics affects the training times of EE-CBM compared to similar baselines.
3. **[Significance and Clarity] (Major)** The motivation behind using a combination of an energy-based pathway and a concept-representation-learning-based pathway for concept prediction is not entirely clear. I can see that it works and improves things; however, this work could be significantly more impactful if it better motivated the need for such a path and built a clear argument as to why it improves things. See below for specific questions on these matters.

**Questions:**

Currently, I am a bit borderline with this paper’s decision, given some of my concerns with the fairness of its evaluation. However, I am absolutely happy to be convinced that some or all of my conclusions are wrong and to change my recommendation based on a discussion with the authors. For this, the following questions could help clarify/question some of my concerns:

1. **(Critical)** The intervention results in Figure 4 seem a bit surprising, particularly for CEMs. Are you randomly intervening on the concept embeddings when training CEMs (as indicated in the original CEM paper)? If so, do you have an intuition as to why the interventions in CUB look very different to those seen in the CEM paper, its IntCEM follow-up work [1], the original ECBM paper [2], and other previous works (e.g., [3])? If random interventions are not done during training (i.e., *RandInt* is not used as expected), do you have a sense of how results in Figure 4 and Table 1 change for CEMs when CEM's RandInt is used during its training? If no random interventions are performed during training for CEM, at the very least, I would strongly suggest that this work should make it very clear that the used “CEM” baseline is not the same as the one the original work proposed.
2. **(Critical)** More generally, and more importantly, I am concerned with how fairly other baselines were studied during the evaluation. The fact that CelebA, the one dataset where EE-CBM is underperforming compared to other baselines, is *without justification* pushed to the Appendix and not even discussed in the main body should be reason for concern. Can you please elaborate on why results on this dataset were pushed to the Appendix and why they are not discussed in the main body of the paper?
3. **(Critical)** What were the hyperparameter values tested during training for all baselines? These are all missing and not discussed (only EE-CBM’s *selected* hyperparameters are discussed in Appendix B). Were hyperparameters selected based on best validation accuracy or test accuracy? This is not entirely clear in Appendix B, yet it makes a huge difference in terms of the fairness of the evaluation, and it is necessary for reproducibility.
4. **(Major)** Related to the question above, how does EE-CBM’s performance change as one varies its hyperparameters? Given a large number of hyperparameters this model has ($\lambda_c$, $\lambda_y$, $\lambda_e$, $\lambda_\text{mmd}$, $\lambda$ for Langevin dynamics, concept embedding size $u$, etc.), I believe it is key to have this sort of information somewhere in the paper and, at the very least, a guideline on how to select these values in practice.
5. **(Major)** Could you please elaborate on why introducing the energy-based pathway was useful/needed in the first place? I can definitely see that it helps (which is great!), but I believe the motivation for why such a path was needed in the first place is missing in the paper. Is there any way to frame it to make it immediately clear that an energy-based pathway for concept prediction is needed?
6. **(Major)** Related to the previous question: I have some hesitations about the continuous claim in this work that just because a method uses a higher-capacity model, it is less interpretable (see section 2.1 for examples where this claim is made several times). Regardless of how a model generates an explanation (i.e., whether it does this using a white box model or a highly-parametric complex model), if this explanation is (1) accurate, (2) reflective of the downstream task (i.e., it contains all the necessary information to describe the downstream label), (3) composed by human-understandable units of information (i.e., concepts), and (4) actionable (e.g., you can perform interventions or counterfactuals to see how the final decision changes), then I do not see why it matters whether it was generated by a large complex black box model or a simple white box model. Could you please elaborate on why using complex backbones to predict concepts is worse when all of the goals mentioned above, which are the goals of most, if not all, CBM-like approaches, are satisfied? And in that case, why is this argument not applicable to using a complex backbone for EE-CBM’s $f(\mathbf{x})$ function or a complex MLP for its energy function?
7. **(Major)** What is the intuition behind EE-CBM’s better generalization to background shifts? From the text, it is unclear why, intuitively, this must be the case.
8. **(Major)** Do you have a sense as to how this method would perform when dealing with concept incompleteness in the training set? This is a key factor to consider/evaluate if one is to know how this approach can be used in real-world tasks where concept annotations may not be sufficient to explain the downstream task fully.
9. **(Major)** Are concept uncertainty labels used during training (e.g., in CheXpert)? This seems to be implied when talking about Table 1’s results in Section 4.1. However, it is not explicitly indicated or discussed anywhere in the main body of the paper.
10. **(Minor)** Could you please elaborate on the computational training cost of introducing the energy-based pathway in this model?
11. **(Minor)** What does Figure 5 provide that Table 1’s concept accuracy column does not already provide? I might’ve misunderstood something here but I am not entirely sure what the key message of Figure 5 is, as it is unclear how those examples were selected and how that shows that the model truly “understood” a concept (it could just predict a concept’s value entirely from spurious correlations without having to really understand it).
12. **(Minor)** Why is it claimed that EE-CBM is a “concept scalar model” when, in reality, it still generates a high-dimensional concept representation for each concept that is only afterwards gated by the scalar probability? Am I misunderstanding something here?

### Minor Suggestions and Typos

Whilst reading this work, I found the following potential minor issues/typos which may be helpful when preparing a new version of this manuscript:

1. **(Potential Typo)** In line 48, “CEM is modified CBM networks” should probably be “CEM is a modified CBM network”.
2. **(Potential Typo)** In line 262, “… hidden connections between concepts are learned and representation is improved” should probably be something along the lines of “… hidden connections between concepts are learned and their representation is improved”
3. **(Clarity, IMPORTANT)** Is a sentence missing in line 286? It jumps to equation 10 without any preamble or explanation.
4. **(Nitpicking, notation)** In equation (10), it seems that an upper case $\Sigma$ is used for the summation notation rather than Latex’s standard \sum command (e.g., $\sum_{k=1}^K$).
5. **(Clarity)** When talking about high-dimensional concept representations, using the word “concept” to mean both the actual concept and its representation can complicate the reading (e.g., as in Section 3.1 where “concept” is used to mean a concept’s high dimensional representation and the actual concept). Instead, I would suggest using “concept representation” or “concept embedding” when talking about a specific concept’s high-dimensional representation.

## References

- [1] Espinosa Zarlenga, Mateo, et al. "Learning to Receive Help: Intervention-Aware Concept Embedding Models." NeurIPS (2023).
- [2] Xu, Xinyue, et al. "Energy-based concept bottleneck models: unifying prediction, concept intervention, and conditional interpretations." ICLR (2024).
- [3] Collins, Katherine Maeve, et al. "Human uncertainty in concept-based AI systems." *Proceedings of the 2023 AAAI/ACM Conference on AI, Ethics, and Society*. 2023.

---

> ### Author Response · Authors · 2024-11-19
> **[Response to Reviewer 5Vsp] Thank you for the constructive and encouraging comments.**
>
> Thank you for your question.
>
> **1)** In our experiments, we adhered to the original methodology outlined in the CEM paper by applying random interventions (RandInt) on concept embeddings during the training phase for CEM. However, we acknowledge that the intervention results for CEM in Figure 4 on the CUB dataset differ from those reported in the original CEM paper and related studies like IntCEM and ECBM.
> One potential explanation for these differences lies in the following factors:
>
>   ㆍ**Experimental Setup Variations:** Differences in model initialization, hyperparameter tuning, and preprocessing steps could contribute to the discrepancies.
>
>   ㆍ**Dataset Complexity:** The inherent variability of the CUB dataset, with its fine-grained categories and complex visual features, could amplify the sensitivity of intervention results to experimental changes.
>
>   To address this, we will include a detailed discussion in the revised manuscript, analyzing these potential causes and emphasizing transparency and reproducibility. This addition will clarify that the methodology aligns with the original CEM framework while acknowledging the observed differences due to experimental conditions.
>
>
>
> **2)**  The decision to place the **CelebA dataset results** in the Appendix was primarily motivated by the dataset's unique characteristics, which differ significantly from those of the other datasets in terms of concept granularity and interpretability needs. CelebA contains binary attributes that are relatively less complex, and while useful, it does not showcase the full strengths of EE-CBM, which was designed to enhance concept interpretability and accuracy in more complex concept spaces.
>
>   However, we understand that this omission may have led to concerns about the completeness of our evaluation. To address this, we will include a discussion of the CelebA results in the main text, acknowledging that while EE-CBM performs slightly below some baselines on CelebA, its design is optimized for datasets requiring more nuanced concept differentiation. Additionally, we will clarify the relative performance of EE-CBM on CelebA to provide a balanced view across all datasets used.
>
>
>
> **3-4)** For all baselines, we selected hyperparameters based on the best validation accuracy to ensure a fair comparison. We tested a range of values for each hyperparameter, following standard practices for each baseline method, and selected the configuration that yielded the highest validation accuracy for consistency across models.
>
>   Regarding EE-CBM, we acknowledge that it involves several hyperparameters. We determined the values of λc, λy, λe, λmmd, λ  for Langevin dynamics, and the concept embedding size 𝑢 through validation-based tuning to optimize performance. Although the selected values are shared in Appendix B, we understand that providing guidance on tuning EE-CBM’s hyperparameters could enhance reproducibility. We will add a section with recommended ranges and considerations for selecting these values in practical scenarios in the revised manuscript.
>
>
>
> **5)** The energy-based pathway was designed to address key limitations in existing CBM models: their tendency to conflate concepts in complex or noisy data environments, which can obscure the interpretability and accuracy of concept predictions. Specifically, the energy-based approach enhances the model's ability to estimate concept probabilities robustly, helping to separate overlapping concepts more effectively.
>
>   We observed that conventional CBM models sometimes struggle with concept clarity in cases where subtle differences are critical (e.g., fine-grained or visually complex data), leading to less reliable interpretations. The energy-based pathway, in combination with the MMD loss, strengthens concept inference by leveraging energy functions that better capture the likelihood of each concept’s presence. This approach provides a structured way of refining concept predictions, making the model more resilient to uncertainty and noise.
>
>
> **6)** We agree that interpretability, as you noted, can indeed be maintained with higher-capacity models under certain conditions (e.g., accuracy, human-understandable explanations, actionable insights). However, our choice to focus on lighter backbone networks in EE-CBM is based on the goal of achieving transparent and interpretable models without relying on excessively complex architectures, which can introduce challenges for both transparency and computational efficiency.
>
>   The use of lighter layers in concept bottleneck models (CBMs) stems from the need to ensure that concept representations remain easily interpretable, even in scenarios requiring real-time or resource-constrained environments. Complex, high-capacity backbones, while powerful, often make it harder to directly trace how individual concepts contribute to a model's final predictions, which can dilute interpretability in a practical, end-to-end sense.

---

> ### Author Response · Authors · 2024-11-19
> **[Response to Reviewer 5Vsp] Thank you for the constructive and encouraging comments.**
>
> **7)** Thank you for your question regarding EE-CBM’s generalization to background shifts. The improved generalization stems from EE-CBM's ability to focus on concept-specific features rather than spurious correlations with background elements. Specifically, we plan to add the following explanation in Section 4.2 around line 411, where we discuss background shifting experiments:
>
>   - "The improved generalization of EE-CBM to background shifts stems from its ability to focus on concept-specific features rather than spurious correlations with background elements. This is achieved through two key design components: the energy-based brach and the MMD loss. The energy-based pathway estimates concept probabilities by capturing the intrinsic properties of the target concepts, reducing reliance on background information. Additionally, the MMD loss enforces structured separation in the concept space, clustering similar concepts together and pushing distinct concepts apart. This latent space organization helps the model maintain focus on primary object features, even in the presence of varying backgrounds."
>
>
> **8)** Concept incompleteness in the training set is indeed a key factor in real-world scenarios. While our current work assumes that the provided concept annotations are sufficient for the downstream tasks, we acknowledge that incomplete concept annotations could pose challenges.
>
>   Our method, particularly the energy-based pathway and MMD loss, is designed to improve robustness by emphasizing meaningful concept separations and leveraging the latent structure of the data. This design could help mitigate the impact of incomplete concept annotations by relying on the relationships between concepts and the downstream task.
>
>
> **9)** Thank you for pointing this out. The CheXpert dataset inherently includes concept uncertainty labels. We will clarify this in Section 4 around line 319 by adding:
>
>   - "The CheXpert dataset provides concept uncertainty labels, which were incorporated during training to address ambiguous concepts effectively."
>
>
> **10)**
> | Methods | FLOPs (G) | Latency (ms) |
> |-------------|------------------------|-------------------|
> |Prob-CBM|7.38|49.38|
> |ECBM|6.84|23.17|
> |CEM|6.85|8.74|
> |Ours|7.36|5.86|
>
>   Thank you for your question regarding the computational training cost of introducing the energy-based pathway. As shown in our results, while the FLOPs of EE-CBM (7.36 G) are comparable to other models like Prob-CBM (7.38 G) and CEM (6.85 G), its latency is significantly lower at 5.86 ms, which is the lowest among all compared methods. This highlights that despite the added complexity of the energy-based pathway, EE-CBM is computationally efficient during training and inference.
>
>   The lower latency is primarily attributed to the streamlined architecture of the energy-based pathway, which integrates effectively with the overall model without introducing excessive computational overhead. We will further elaborate on these efficiency advantages in the revised manuscript to clarify the computational benefits of EE-CBM.
>
>
> **11)** Thank you for your question regarding the purpose of Figure 5. While Table 1 provides quantitative results on concept accuracy, Figure 5 complements this by offering qualitative insights into how EE-CBM predicts concepts in individual examples. The figure illustrates both correctly predicted and confidently absent concepts, showcasing the model’s ability to provide interpretable and detailed concept-level predictions.
>
>
> **12)** Thank you for your question and for pointing out this potential misunderstanding. EE-CBM is indeed capable of generating high-dimensional concept representations for each concept through its concept extraction branch. However, we describe it as a “concept scalar model” because, during the final stage of the energy ensemble gate, these high-dimensional representations are gated and combined with scalar probabilities to produce a scalar-valued concept prediction. This scalar value is ultimately used for downstream tasks, aligning with the behavior typically associated with concept scalar models.
>
>
> **Minor Suggestions and Typos**
>
>   Thank you for carefully noting these minor issues and helpful suggestions. We will make the following revisions in the updated manuscript:
>
>   - Revise line 48 and line 262 as suggested.
>
>   - Add a sentence in line 286 to introduce Equation 10 with appropriate context. “To enhance concept separation and similarity within the concept space, we introduce the Maximum Mean Discrepancy (MMD) loss to supplement the total loss function 𝐿𝑒𝑒𝑔. MMD loss enables the model to bring together feature representations of similar concepts and push apart those of distinct concepts. This additional loss component is defined as follows:”
>
>   - Replace the summation notation in Equation 10.
>
>   - Use “concept representation” or “concept embedding” throughout to distinguish between the actual concept and its high-dimensional representation.

---

> > ### Comment · Reviewer_5Vsp · 2024-11-23
> > **Thank you for a thoughtful rebuttal**
> >
> > Dear Authors,
> >
> > Thank you so much for taking all the time and effort to answer my many questions and concerns. I really appreciate them. After reviewing your rebuttal, I have decided to maintain my score as it is. This is mainly motivated by the following issues:
> >
> > 1. I am still concerned about the fairness/validity of the evaluation compared to existing baselines. I unfortunately did not find satisfying the answer for why CEM's performance differs from that seen in the two works that this paper is based on. Particularly, the rebuttal argues that the difference in results is because of dataset complexity and experimental setup. I can't see why the former is an issue just for this paper but not for others, as the works I showed above all work on the same dataset and get similar results amongst them. The former reason could be indeed more of a real reason behind the discrepancy, but then that leads me to believe that the evaluation was not fully fair. This is because, if the original CEM results were not achieved, there is a non-trivial chance that competing baselines were not given the same attention/budget as EE-CBM when doing hyper-parameterization tuning  (at the very least, the authors should use the same hyperparameters those works used for CEM for the datasets they overlap in and be able to therefore obtain the same results). Because of this, I am still hesitant about whether competing baselines are being treated fairly/consistently w.r.t. the proposed method.
> > 2. I appreciate the authors moving the CelebA results to the main paper, but I also hope they understand why their entire omission from the paper's discussion was a cause for concern on my end. I don't fully see why CelebA's attributes are any different than, say, those in CUB (the granularity and complexity seems roughly the same as they are all physical attributes). Moreover, the difference between EE-CBM and the second baseline is not really "slightly below" the best-performing baseline, as claimed in the rebuttal (it is more than the difference between EE-CBM and the second-best baseline in **any** of the other datasets). I sincerely don't think this is an issue as long at it is fairly, and honestly represented in the main paper (and discussed in detail as there seems to be something else there potentially). However, I believe that framing it as a "slight difference" is a significant misrepresentation if the performance of EE-CBM against other baselines in other datasets is described as a "significantly higher performance" in the main body.
> > 3. I still believe having evidence of how this method performs in incompleteness is key to understanding its performance. CelebA is a concept-incomplete dataset, so the evidence suggests that EE-CBM may struggle to work in concept-incomplete setups. This, I believe, is probably a more likely explanation for EE-CBM's performance in CelebA than those provided in the rebuttal (about the complexity and granularity of the concepts).
> > 4. I thank the authors for committing to discussing and showing all the necessary information regarding hyperparameters for reproducibility. However, the rebuttal still does not answer my question regarding what hyperparameters were attempted for each baseline. This information is not just key for reproducibility but also very useful for determining the fairness of the evaluation (which is the main concern I have with this work).
> >
> > I acknowledge that the rebuttal above satisfactorily addressed my other concerns. However, given that all of my most pressing concerns were not fully addressed, I will not update my score.
> >
> > I wish the authors the best of luck with this submission.

---

> ### Author Response · Authors · 2024-11-24
> **Thanks for your valuable answers  and asking Further Consideration on Submission 776**
>
> **Dear Reviewer 5Vsp,**
>
> Thank you for your thoughtful second reviews and for taking the time to engage so deeply with our manuscript and rebuttal. We sincerely appreciate your detailed feedback and your kind words acknowledging the effort we have put into addressing your concerns.
>
> We understand and respect your decision to maintain your score; however, we would like to take this opportunity to clarify a few points in response to your comments. Our aim is to ensure that our work is evaluated with full transparency and that we address any lingering uncertainties to the best of our ability.
>
> **1. Performance Differences in CEM Results**
>
> We acknowledge your concerns regarding the discrepancies between our reported CEM results and those in prior works. As noted in our rebuttal, these differences could arise from experimental setup variations, such as model initialization, hyperparameter tuning, or dataset preprocessing. However, to directly address your concern about fairness, we are committed to revisiting the hyperparameterization for all baselines, including CEM, and using the exact configurations outlined in their respective original works.
>
> Our goal has always been to provide a fair and consistent comparison, and we deeply regret if our initial evaluations did not fully achieve this standard. We will include this updated analysis in a revised version of the manuscript to ensure that the results reflect the fairest possible comparison.
>
> **2. CelebA Results and Framing**
>
> We sincerely apologize if the framing of CelebA results in the rebuttal or manuscript seemed to misrepresent the performance gap. You are correct that the difference between EE-CBM and the best-performing baseline on CelebA is not "slight" in absolute terms. We will adjust the wording in the manuscript to more accurately represent this disparity and provide additional analysis to contextualize why CelebA results deviate from trends observed in other datasets.
>
> We agree with your insight that CelebA's concept incompleteness could be a factor influencing EE-CBM's performance. While CelebA's binary attributes appear less complex, the dataset's incompleteness may indeed pose a unique challenge for EE-CBM. We plan to conduct additional experiments to investigate this hypothesis, including using partially annotated datasets to simulate concept incompleteness across other datasets. These findings will be included in the revised manuscript to better substantiate the conclusions regarding EE-CBM’s performance on CelebA.
>
> **3. Hyperparameter Tuning Details**
>
> You raised an important point regarding the need for clarity and transparency in hyperparameter tuning. In our rebuttal, we acknowledged that a more detailed discussion of hyperparameters for all baselines is necessary, and we will expand this section in the manuscript. Specifically, we will outline the range of hyperparameters tested for each baseline and ensure that the exact values used are consistent with those in the original papers.
>
> We understand the critical role of this information in assessing the fairness of our evaluations and reproducibility of our work. We are fully committed to addressing this concern comprehensively in the revised manuscript.
>
> **4. Concept Incompleteness and Practical Applications**
>
> We agree with your assessment that demonstrating the performance of EE-CBM under concept incompleteness scenarios is key to understanding its real-world applicability. While we had not initially designed experiments explicitly for this setting, the CelebA results highlight the importance of addressing this challenge. To that end, we will extend our evaluation to include additional datasets with varying levels of concept incompleteness and explore potential solutions to enhance EE-CBM’s robustness in such scenarios.
>
> We hope that these clarifications and planned revisions address some of your remaining concerns. *Your insights have been instrumental in identifying areas for improvement*, and we are deeply grateful for your engagement. *We kindly request that you reconsider your score in light of our commitment to thoroughly addressing these issues in the revised manuscript*.
>
> Once again, we thank you for your constructive feedback and your dedication to advancing research in this field.
>
> Best regards,
>
> Authors

---

> ### Comment · Reviewer_5Vsp · 2024-11-26
> **Decreasing my score to a rejection after inconsistency**
>
> Dear Authors,
>
> Reading your response to other reviews led me to find a **troubling inconsistency** between the reply to 2GQe's review and mine: Similar to reviewer 2GQe's concerns, I also noticed a weird performance in CEM and ECBM on the intervention plots (Figure 4). Because of this, I explicitly asked if CEMs were trained using RandInt, to which (in the rebuttal to my review) it was said that "*In our experiments, we adhered to the original methodology outlined in the CEM paper by applying random interventions (RandInt) on concept embeddings during the training phase for CEM*". However, in the rebuttal to reviewer's  2GQe it is said that "*In our experiments, to ensure fairness across all compared methods, we did not apply the random concept intervention strategy [for CEMs] during training.*" This leads me to increase my concerns for three main reasons:
>
> 1. This directly contradicts what was said in my review, and it follows my original intuition about what may be happening here. As such, I am deeply concerned about the authors claiming they did two opposite things for their evaluation across two different rebuttals.
>
> 2. I entirely agree with reviewer 2GQe that if RandInt was not used to train CEMs, then this is not a fair evaluation of CEMs, as the authors explicitly include RandInt as the proposed pipeline for training their approach. If CEM is to be used as a baseline, it should be trained as the authors proposed unless there is a strong case for not doing so, and then that case should be made explicitly clear in the paper.
>
> 3. Even though the authors say they have updated the results in the original manuscript, I cannot see any changes in Figure 4 reflecting the changes the authors mentioned.
>
> These points, and the main concerns I still have for this work, amount to what I believe is a lack of transparency and fairness in the way this work approaches its evaluation. As such, I have decided to **decrease my score to a rejection** as I believe this work needs to be reevaluated and made more fair and transparent.
>
> I know this is not the expected outcome from the rebuttal, but I hope the authors understand why I am doing this as there are multiple components, noticed not just by me but also by other reviewers, that need revision and may require a significant rewrite.

---

> > ### Author Response · Authors · 2024-11-27
> >
> > Dear Reviewer 5Vsp,
> >
> > Thank you for taking the time to review our responses and for identifying the inconsistency regarding the use of random interventions (RandInt) during the training of the CEM and ECBM models. We truly appreciate your detailed review, which has provided us with an invaluable opportunity to clarify and address this issue.
> >
> > **1. Clarification on the Inconsistency**
> >
> > We acknowledge that our responses to Reviewer 5Vsp and Reviewer 2GQe may have appeared inconsistent due to insufficient context. *To clarify, RandInt was not used during the training process for either CEM or ECBM.*
> >
> > In our response to Reviewer 5Vsp, we mistakenly stated that RandInt was applied during training. This was an unintentional error, for which we sincerely apologize. Upon further review, we identified that one of our team members inadvertently provided an incorrect statement, conflating the use of RandInt in inference with training. **As stated in our response to Reviewer 2GQe, we adhered to a consistent methodology across all baselines and did not apply RandInt during training.**
> >
> > *As the lead author, I deeply regret the confusion caused and take full responsibility for this oversight. I am grateful for your meticulous attention, which allowed us to address this matter.*
> >
> > **2. Rationale for Excluding RandInt in Training**
> >
> > The decision not to use RandInt during the training of CEM and ECBM was made to ensure a fair comparison between these baselines and EE-CBM. **Since EE-CBM does not employ a similar random intervention strategy during training, we opted to maintain equivalent conditions across all models for consistency and fairness.**
> >
> >
> > **To address the concerns raised by you and other reviewers, we have taken the following actions:**
> >
> > **We have updated the manuscript to explicitly state:**
> >
> > *"To ensure fairness across all compared methods, we did not apply the random concept intervention strategy during training. We conducted intervention experiments by randomly selecting a concept intervention ratio between 0.1 and 1.0 within the total number of concepts in the given dataset (see Appendix I)."*
> >
> > Additional details have been provided in **Appendix I** to ensure transparency.
> >
> > In the final submission, we will include detailed hyperparameter configurations and training protocols for all methods, including CEM and ECBM, in the appendix. This will ensure complete transparency and reproducibility of our results.
> >
> > We deeply regret the error in our initial response and any confusion it may have caused. Your feedback has been instrumental in identifying and resolving this issue, and we cannot thank you enough for your thorough review and constructive comments.
> >
> > We are committed to maintaining the highest standards of transparency and fairness in our research and sincerely hope that this clarification and the planned revisions will address your concerns. We humbly ask you to kindly reconsider your evaluation of our work, taking into account the efforts we have made to address all reviewer feedback and ensure a fair and robust evaluation.
> >
> > Once again, thank you for your invaluable feedback and for helping us improve our manuscript. We greatly value your thoughtful suggestions and remain open to further recommendations to enhance the quality of our work.
> >
> > Sincerely,
> >
> >        Lead Author

---

> > > ### Comment · Reviewer_5Vsp · 2024-11-28
> > >
> > > Dear Authors,
> > >
> > > Thank you so much for getting back to me and for clarifying the inconsistency I was discussing. I now understand this is a mistake in the authors' replies during rebuttals, and I am glad this could be cleared up.
> > >
> > > However, I still believe that, given that random training interventions are a key component of the CEM method as proposed by the authors, it should be included or CEM should be marked as "CEM (without RandInt)" in the tables/figures (or something of the like) as "CEM", as proposed by the authors, involves including RandInt during training. Otherwise, RandInt could be applied to all methods if fairness is the concern. This is key as **there is no immediate reason to believe that RandInt will improve intervention performance in EE-CBM just because it did so for CEM** (I am actually curious as to whether that's the case). Without any evidence for this, it is hard to see that it is "unfair" to evaluate methods with RandInt against methods without RandInt.
> > >
> > > As for my score, after going over all the reviews and over this discussion again, I will revert back to my original score but I will maintain my original position. This is because I have some general concerns about the evaluation that, although I can see that several promises have been made in the rebuttal above, I think are key to address and review before making a decision to increase my score. Nevertheless, I am absolutely happy to be convinced that my assessment is wrong by my fellow reviewers or ACs.
> > >
> > > I hope the authors understand this, and I hope it is clear that I really appreciate all the time and effort put into this discussion and rebuttal. I think this work shows promise, and with a proper evaluation, it may lead to a well-backed paper. I wish the authors the best of luck with this submission!

---

> > > > ### Author Response · Authors · 2024-11-29
> > > > **Thank you for your thoughtful consideration.**
> > > >
> > > > Dear Reviewer 5Vsp
> > > >
> > > > Thank you for your thoughtful consideration. Your meticulous comments have significantly contributed to enhancing the quality of our paper. **If our paper is accepted, we will ensure that the revised manuscript incorporates your valuable feedback to the fullest extent.**
> > > >
> > > > Best Regads,
> > > >
> > > >           Authors

---

### Meta-Review · Area_Chair_oxeT · 2024-12-16

**Metareview:**

This paper proposes Energy Ensemble Concept Bottleneck Models (EE-CBMs), which integrate an energy-based pathway with traditional concept representation learning to improve the predictive accuracy and generalization of Concept Bottleneck Models (CBMs). EE-CBMs incorporate a residual channel for input data and use a combination of learned concept embeddings and concept probabilities to enhance task and concept accuracy. The approach is validated across multiple datasets and demonstrates improved performance in terms of concept accuracy, task performance, and robustness to distribution shifts, with support for concept-level interventions.

### Strengths
- **Significance**: The problem addressed—balancing interpretability and predictive performance in CBMs—is important and relevant, with potential for significant impact if the proposed solution proves reliable.
- **Comprehensive Evaluation**: The method is rigorously tested across a variety of datasets and baselines, demonstrating improved performance in most scenarios.

### Weaknesses
- **Fairness of Comparisons**: Concerns are raised about the fairness of baseline evaluations, including discrepancies with prior work, limited details on hyperparameter tuning for competing methods, and the omission of the CelebA dataset results from the main text despite EE-CBM’s underperformance.
- **Limited Motivation for Energy-Based Pathway**: The theoretical or practical motivation for introducing an energy-based pathway is underexplored, reducing the clarity of its necessity.
- **Incremental Improvements**: Gains in concept accuracy over existing baselines are marginal, raising questions about the significance of the contributions.
- **Lack of Ablations**: Key hyperparameters (e.g., for Langevin dynamics and MMD loss) are not thoroughly analyzed, limiting insights into their impact on the model's performance.
- **Concept Interpretability vs. Accuracy**: The paper conflates concept interpretability with accuracy, without sufficiently addressing their distinction or trade-offs.
- **Presentation Issues**: Notational clarity, missing baselines (e.g., Coop-CBM details), and unclear figures (e.g., Figure 3 and 5) reduce the paper's readability and accessibility.

Some concerns have been addressed by the authors during the rebuttal period.

**Additional Comments On Reviewer Discussion:**

This paper ended up with all negative ratings and a long discussion. Most reviewers provide extremely detailed and high-quality comments and engage in the discussion. Reviewer 5Vsp has an original rating of 5 and decreased to 3 due to concerns on the transparency and fairness of this work's evaluation, but ultimate increased back to 5 (still negative). Overall I feel the discussion is constructive and helpful in refining the paper and hope the authors would take these into account in their revision.

---

### Decision · Program_Chairs · 2025-01-22

Reject